# Home-Based Music Therapy to Support Bulbar and Respiratory Functions of Persons with Early and Mid-Stage Amyotrophic Lateral Sclerosis—Protocol and Results from a Feasibility Study

**DOI:** 10.3390/brainsci12040494

**Published:** 2022-04-13

**Authors:** Alisa T. Apreleva Kolomeytseva, Lev Brylev, Marziye Eshghi, Zhanna Bottaeva, Jufen Zhang, Jörg C. Fachner, Alexander J. Street

**Affiliations:** 1ALSMusicTherapy.Org, 115419 Moscow, Russia; muzterapevt@gmail.com; 2Bujanov Moscow City Clinical Hospital, 115419 Moscow, Russia; lev.brylev@gmail.com; 3Institute of Higher Nervous Activity and Neurophysiology, 115419 Moscow, Russia; 4Moscow Research and Clinical Center for Neuropsychiatry, 115419 Moscow, Russia; 5Department of Communication Sciences and Disorders, MGH Institute of Health Professions, Boston, MA 02129-4557, USA; marziye_eshghi@med.unc.edu; 6Clinical and Research Institute of Emergency Pediatric Surgery and Trauma, 119180 Moscow, Russia; zhanna.s.bottaeva@gmail.com; 7Faculty of Health, Education, Medicine & Social Care, School of Medicine, Anglia Ruskin University, Cambridge CM1 1SQ, UK; jufen.zhang@aru.ac.uk; 8Music, Health and the Brain, Cambridge Institute for Music Therapy Research, Anglia Ruskin University, Cambridge CM1 1SQ, UK; jorg.fachner@aru.ac.uk

**Keywords:** amyotrophic lateral sclerosis (ALS), motor neurone disease (MND), bulbar, respiratory training, speech, swallowing, cough, communication, music therapy, music, rehabilitation, palliative

## Abstract

Respiratory failure, malnutrition, aspiration pneumonia, and dehydration are the precursors to mortality in ALS. Loss of natural communication is considered one of the worst aspects of ALS. This first study to test the feasibility of a music therapy protocol for bulbar and respiratory rehabilitation in ALS employs a mixed-methods case study series design with repeated measures. Newly diagnosed patients meeting the inclusion criteria were invited to participate, until the desired sample size (*n* = 8) was achieved. The protocol was delivered to participants in their homes twice weekly for six weeks. Individualised exercise sets for independent practice were provided. Feasibility data (recruitment, retention, adherence, tolerability, self-motivation and personal impressions) were collected. Bulbar and respiratory changes were objectively measured. Results. A high recruitment rate (100%), a high retention rate (87.5%) and high mean adherence to treatment (95.4%) provide evidence for the feasibility of the study protocol. The treatment was well tolerated. Mean adherence to the suggested independent exercise routine was 53%. The outcome measurements to evaluate the therapy-induced change in bulbar and respiratory functions were defined. Findings suggest that the protocol is safe to use in early- and mid-stage ALS and that music therapy was beneficial for the participants’ bulbar and respiratory functions. Mean trends suggesting that these functions were sustained or improved during the treatment period were observed for most outcome parameters: Maximal Inspiratory Pressure, Maximal Expiratory Pressure, Peak Expiratory Flow, the Center for Neurologic Study—Bulbar Function Scale speech and swallowing subscales, Maximum Phonation Time, Maximum Repetition Rate—Alternating, Maximum Repetition Rate—Sequential, Jitter, Shimmer, NHR, Speaking rate, Speech–pause ratio, Pause frequency, hypernasality level, Time-to-Laryngeal Vestibule Closure, Maximum Pharyngeal Constriction Area, Peak Position of the Hyoid Bone, Total Pharyngeal Residue C24area. Conclusion. The suggested design and protocol are feasible for a larger study, with some modifications, including aerodynamic measure of nasalance, abbreviated voice sampling and psychological screening.

## 1. Introduction

Amyotrophic lateral sclerosis (ALS) is a rapidly progressive fatal neurological disease, affecting motoneurons in the brain and spinal cord. According to a meta-analysis, the overall pooled incidence of ALS was 1.68 (95% confidence interval, CI 1.50–1.85) per 100,000 person-years [1], and prevalence is increasing [2]. Clinical presentation is phenotypically heterogeneous and depends on the site of onset. Limb onset is the most common and the first symptoms include compromised gait and manual dexterity. Bulbar-onset ALS patients first develop symptoms in the head and neck region, experiencing slurred speech or difficulty swallowing. There are also rare truncal–abdominal (axial) and respiratory onsets. At least 90% of ALS cases are considered sporadic, which means the disease seems to occur with no family history of the disease [3]. The pathophysiology mechanisms behind ALS are not clear and may include oxidative stress, protein misfolding and aggregation, skeletal muscle dysfunction, glutamate excitotoxicity, mitochondrial dysfunction, neuroinflammation, RNA processing disruption, and apoptosis [4].

### 1.1. Bulbar and Respiratory Involvement in ALS

A total of 93% of persons with ALS (pwALS) experience speech impairments and 85% present with swallowing dysfunction at some point during disease progression [5]. Loss of natural communication is regarded by patients as one of the worst aspects of the disease [6]. Patients with dysphagia report social isolation, fear and decreased mental health [7]. Dysphagia and malnutrition contribute to 25.9% of ALS mortality and increase the risk of death by 7.7 fold [8,9,10]. Aspiration pneumonia and dehydration have also been cited among the leading factors contributing to mortality in ALS [11,12]. Emotional lability (pseudobulbar affect), a symptom frequently correlated with bulbar involvement in ALS, may also be confusing and disruptive, especially when communicating with those who are not aware of the nature of the problem [13]. Respiratory dysfunction, manifested as dyspnoea on exertion, orthopnoea, and early morning headaches with fatigue [14], is common in ALS. Weakening of respiratory function and adherence to NIV are the leading causes of anxiety of pwALS [15,16], and respiratory failure has been cited as the leading cause of death [17].

### 1.2. Cognitive, Behavioural and Psychological Symptoms in ALS

Whilst in the past ALS was considered distinctly a disorder of the motor system, current evidence suggests that some cognitive impairment (ALSci) or behavioural impairment (ALSbi) occurs in up to 50% of cases, and co-morbid dementia (ALS-FTD) occurs in approximately 14% of patients with a new diagnosis of ALS [18]. The notion that “pure” ALS and “pure” frontotemporal dementia (FTD) may present two extremes of one disease continuum [19] is reinforced by identification of transactive response DNA-binding protein 43 (TDP-43) as a major pathological substrate underlying both diseases [20]. ALSci and ALS-FTD patients are more severely impaired in executive function, attention, language and memory than cognitively intact ALS patients.

Depression is prevalent in ALS. It is associated with disease severity at initial assessment and has detrimental effects on survival and quality of life [21]. It is reported that high levels of anxiety are often present in persons with ALS during the diagnostic phase and the earlier period after the diagnosis [22]. Depression and anxiety are undertreated in ALS, and psychological and pharmacological interventions for their prevention and treatment are only minimally addressed in the literature [23].

### 1.3. Family Caregivers in ALS Care

Family caregivers play a key role in ALS care, providing care for 47 h per week on average [24] and actively participating in clinical decision making [25]. Nearly all treatment or rehabilitative activity intended for persons with ALS requires caregivers’ participation. Caregivers’ physical and emotional health and quality of life are significantly impacted when ALS occurs in the family [26,27]. Including caregiver’s perspectives is important when designing and researching complementary ALS treatments. 

### 1.4. The Role of Exercise

Non-invasive ventilation (NIV) for respiratory support, gastrostomy feeding for dysphagia, riluzole and edaravone medications are the four medical methods known to prolong survival for pwALS [28,29].

Until recently, pwALS have been discouraged from exercising based on the assumption that exercise can lead to muscle fatigue and increase patients’ disability [30]. However, this assumption is not supported by the research [31,32,33,34]. To the contrary, there is evidence that physical inactivity secondary to ALS may lead to cardiovascular deconditioning, disuse weakness and consequential muscle atrophy. Some reviews [35,36] support evidence for moderate physical exercise in ALS, whilst highlighting the importance of supervised, individualised training programs. One Cochrane review [37] concluded there was no solid evidence to deem exercise in ALS beneficial or harmful and emphasised the need for further research. A literature review [5] examined a total of 18 exercise-based intervention studies published in peer-reviewed journals between 1960 and 2014. Of these, no experimental studies examined the impact of targeted exercise on speech or swallowing function. Mild- to moderate-intensity limb or respiratory exercise, applied early in the disease, was noted to have a beneficial impact on motor function and survival. New studies [38,39] suggest that strictly monitored exercise programs reduce motor deterioration and improve the functional independence of pwALS. Although there is a lack of evidence supporting the use of strengthening exercises for improving speech in pwALS, there is no evidence of such exercises being harmful [11]. Emerging evidence suggests that respiratory training may have a positive effect on respiratory and swallowing functions in ALS [40,41,42]. 

In the absence of a curative treatment, a wider range of rehabilitative options has to be considered for pwALS, enabling them to reach their fullest potential and best quality of life (QoL), delaying disease progression and prolonging lifespan [43].

### 1.5. Music Therapy for ALS: Developing Clinical Approach

Music therapy (MT) is the evidence-based professional clinical application of music and its elements to improve the psychological, emotional, cognitive, physical and social health and well-being of individuals and communities [44,45,46,47]. Professionally trained music therapists are well equipped to provide symptomatic care for people with neurodegenerative diseases [48], adapting interventions to the increasing and changing disability of each patient as the disease progresses, whilst maintaining and developing a trusting therapeutic relationship early in the disease course. Music stimulates global brain activation and shares processing components with other functions, such as those involved in language, movement, reasoning and experiencing emotions [49,50,51,52]. The growth of scientific knowledge about music perception and production and the effects of these on non-musical brain and behaviour functions has led to the development of clinical techniques to treat cognitive, sensory, and motor dysfunctions resulting from neurological injury or disease [53]. 

Research on clinical music therapy applications for ALS is limited [54] and presents significant challenges for an investigator due to the heterogeneous presentation and progression of the disease [55]. There is also a poor scientific understanding of the disease mechanisms [56] and the ethical issues of research involving terminally ill people [57]. It has been suggested that MT could contribute to interdisciplinary ALS care [58]. There is anecdotal evidence that MT increases mind–body connection [59], and reduces distressing physical symptoms such as dyspnoea and pain, as well as associated feelings of loneliness, anxiety and sadness for patients with advanced ALS [60]. It is further suggested that MT is “pleasant and restorative” for pwALS who have tracheostomy and for their families [61]. It provides families affected by ALS with opportunities for shared meaningful activities [62]. The first RCT of MT with ALS found that active MT increased communication, improved QoL and decreased the physical symptoms of the disease for participants during hospital stay [63]. Music-assisted relaxation may be a useful strategy to optimize the non-invasive ventilation experience for pwALS [64]. Albeit currently underused, MT could be one of the modalities of supportive rehabilitation in ALS, potentially providing multiple benefits for pwALS and their families.

In terms of physiological location and neuronal activation, swallowing, vocalisation and breathing are tightly coordinated and closely related processes [65]. No published research addressing the use of MT techniques for the neurorehabilitation of pwALS has been found (e.g., supporting motor, cognitive, respiratory, swallowing, and speech functions), even though empirical evidence and research with other clinical populations suggest that such interventions may be beneficial [66,67,68,69,70]. The patient’s motivation to work towards therapeutic goals may increase with the use of music [53], and since loss of motivation is a major factor in pwALS, this makes music-based interventions worthy of research [71]. 

## 2. Materials and Methods

### 2.1. Study Aim

The aim of this study was to determine the feasibility of a home-based MT protocol as an intervention to support respiratory and bulbar functions in early- and mid-stage ALS. Since this is the first study to examine the role of MT in the physical rehabilitation of pwALS, it has been primarily focused on the safety and tolerability of the treatment protocol. Inquiry was made into the most sensitive and feasible tools to assess the treatment effect on respiratory and bulbar functions in ALS. 

### 2.2. Registration and Ethical Approvals 

This study is registered with the U.S. National Library of Medicine ClinicalTrials.gov database, NCT03604822. Ethics clearance has been obtained from the ethics committee at Anglia Ruskin University (Cambridge, UK; the Arts, Laws and Social Sciences FRDSC meeting on 15 December 2017) and from Moscow Municipal Independent Ethics Committee (Moscow, Russia; decision No. 17/16_14.03.2018).

### 2.3. Study Design

This is an ABA mixed-methods case series with repeated measures. Reliable predictors of decline in ALS are yet to be found [72]. Even the relatively narrow recruitment criteria for this study allowed for vast differences between the participants’ responses to MT treatment and natural disease progression. All participants started the trial simultaneously, each participant receiving treatment and serving as their own control. The duration of participation in this study was 16 weeks, including the lead-in (control) phase (weeks 1–6), the treatment phase (weeks 7–12) and the wash out phase (weeks 13–16) (see Appendix A. Study Consort). Quantitative and qualitative data, including recruitment, retention, adherence, thematic analysis of semi-structured interviews and treatment notes, were collected to determine the feasibility of a home-based MT intervention for bulbar and respiratory rehabilitation in ALS. Biomedical data were collected to assess the long-term and short-term effects of the MT protocol. Such a study design allows for a rigorous and systematic exploration of the processes and outcomes in real-world practice, where the clinical situation is not manipulated [73]. This approach may be especially useful considering the heterogeneity of ALS presentation and progression and the innovative character of the treatment protocol.

### 2.4. Facility

The study host, ALS Centre Moscow (Russia), is a collaboration between three Moscow hospitals that has provided home-based multidisciplinary care for pwALS and their families since 2012. The incidence of ALS in Russia is 1.25 per 100,000 person-years [74]. Approximately 110 ALS patients out of approximately 950 pwALS residing in the Moscow region receive treatment from the team. MT was first introduced at the Centre in 2013. The medical director of the Centre gave full approval for this study. Three research nurses and two research assistants helped to collect data. The principal researcher implemented the six-week home-based MT treatment for this study and coordinated the research team. 

### 2.5. Recruitment

A consecutive sampling strategy was employed in order to avoid bias in regard to disease onset: all newly diagnosed patients living within the Moscow city limits and meeting the inclusion and exclusion criteria were invited to participate. Recruitment continued until the desired sample size (*n* = 8) was achieved or a cut-off date for recruitment was reached. 

### 2.6. Inclusion Criteria

Diagnosis of probable or definite ALS by the revised El Escorial criteria [75] confirmed by the neurologist at ALS Moscow Centre prior to screening for enrolment.
(1)Amyotrophic Lateral Sclerosis Functional Rating Scale—Revised (ALSFRS-R) [76] bulbar subscore ≥ 9, but ≤11, where bulbar subscore = the sum of ALSFRS-R questions 1–3 (maximum score of 12);(2)Forced Vital Capacity (FVC) greater than 60%;(3)Unimpaired cognition as evidenced by Edinburgh Cognitive and Behavioural ALS Screen (ECAS, Russian version) cut-off scores adjusted for age and education [77,78];(4)Able to consent to treatment;(5)Native speaker of Russian.

### 2.7. Exclusion Criteria


(1)Tracheostomy or mechanical ventilation;(2)Diaphragmatic pacer;(3)Significant concurrent respiratory disease;(4)Allergies to barium;(5)Receiving any other experimental treatment for dysarthria, dysphagia, dystussia and dyspnoea for the duration of this study;(6)Receiving any other music therapy treatment for the duration of this study.


### 2.8. Caregivers Participating in This Study

If designated by the primary participants, and only with their permission, their main caregivers were contacted and invited to participate in structured interviews before and after the therapy phase of this study. 

### 2.9. Informed Consent

Participants were provided with a participant information sheet. Each participant, including the participating caregivers, signed voluntary informed consent. Participants were provided with the withdrawal form to sign if they wanted to withdraw from this study at any point. Special care was taken when scheduling the MT treatment to avoid interference with the participant’s daily activities.

### 2.10. The Experimental Music Therapy Treatment Protocol

The experimental music therapy treatment protocol to support bulbar and respiratory functions of persons with early- and mid-stage amyotrophic lateral sclerosis outlines the treatment protocol used in this study. The protocol consists of a series of music-based breathing, gentle stretching, relaxation and singing exercises tailored to the rehabilitative needs of pwALS and constitutes, in a rough approximation, an adaptive voice lesson oriented towards ALD-specific non-musical (rehabilitative) goals. 

Upper and lower motor neurone deterioration in ALS result in dysarthria and dysphagia of spastic-flaccid type, although actual presentation varies in each patient and changes with disease progression. Generally, lower motor neurone involvement, characteristic of bulbar onset, leads to flaccid presentation, whilst degeneration of upper motor neurone, characteristic of spinal onset, results in spasticity of bulbar muscles. Muscle relaxation and moderate exercise, as well as learning efficient breathing and voice production techniques, voice care techniques [79] and ALS-specific communication strategies therefore may be helpful, rather than rigorous strengthening oral motor exercises. Based on literature review, including [5,13,14,40,41,42,80,81,82,83,84,85,86], the following therapy objectives for this protocol were decided upon:(1)To increase breath support,(2)To increase muscle relaxation,(3)To increase speech rate,(4)To prevent or decrease hypernasality, and(5)To maintain coordinated swallow.

Speech-language techniques recommended for pwALS, published MT protocols for bulbar rehabilitation in patients affected by various neurological conditions, were considered [68,87,88,89]. Exercise IX was adapted from Lyle [90]. It involves timed upward-and-backward movement of the soft palate alternated with its relaxation. Proper velopharyngeal closure is essential for healthy swallowing [65] and plays an important role in speech intelligibility by preventing hypernasality [86]. Exercise XII was adapted from a published MT protocol [68] for stroke patients with dysphagia. Vowels ‘a’, ‘I’ and ‘u’ are used because they represent the extremes of vocal tract configuration [91]. Verbal descriptions necessary for understanding the anatomy and physiology of voice production were adopted from a professional voice training technique [92]. Breathing and singing exercises were mostly adopted from McClosky’s vocal technique [93].

The ALS-specific, individualised MT protocol was delivered to the participants in their homes twice weekly for a duration of six weeks by the principal researcher—a professional, board-certified music therapist. Facilitating musical structures were composed by the researcher to support cueing, timing and intensity of breathing and vocalisation exercises. These structures were regularly adjusted by the music therapist to suit the unique capabilities, current individual demands and progress of each participant. One familiar song, selected by each participant upon consultation with the music therapist, was used to facilitate a therapeutic singing exercise closing each session. Caregivers and family members could choose to be present during the sessions, according to the preference of the participating pwALS. 

Printed ALS-specific voice health guidelines, compiled by the researcher and based on general vocal health guidelines and ALS-specific speech-language therapy recommendations (see Appendix B. Voice Care Guidelines for Research Participants—Persons with ALS (English translation)), were provided for participants prior to commencing treatment with the aim of promoting healthy voice use habits in daily life.

Black and white drawings from voice textbooks [92,93], depicting vocal tract, posture, the larynx, the vocal fold, diaphragm action in anterior and lateral view, rib cage and abdominal muscles and balance of the head and neck were used as visual aids during the music therapy sessions. The anatomy and physiology of respiration, voice production and swallowing were briefly explained to participants in order to increase their awareness and sense of control over these processes.

MT constitutes a new treatment modality for bulbar and respiratory dysfunction in ALS; thus, the safety and tolerability of the treatment protocol were prioritised. The protocol allowed for ample rest and relaxation possibilities between short exercises involving active muscle work, in order to prevent voice overuse and fatigue [94]. The participants were advised to wear comfortable clothes allowing for unrestricted movement and breathing. They were encouraged to participate to their comfort level in all the exercises suggested and modelled by the MT, and to pause, rest and hydrate as needed. The duration of each visit was approximately one hour, including session opening and closure. The treatment protocol is outlined in Table 1. Experimental music therapy treatment protocol to support bulbar and respiratory functions of persons with early- and mid-stage amyotrophic lateral sclerosis.

### 2.11. Recommended Exercises for Independent Practice

The participants were encouraged, but not required, to independently practice breathing, relaxation and voice skills learned in MT sessions on the days when no therapist visits were scheduled. A recorded guide (CD or mp3 files provided via USB flash drive) and corresponding printed instructions for independent practice were provided to each participant at the end of the third session (see Appendix C. Sample Recommended Daily Exercises Instructions (English Translation)). The individualised exercise sets consisted of abbreviated (approximately 15 min long) versions of the treatment protocol used in therapy sessions, with adjustments for the tempo, vocal range, preferred musical accompaniment and imagery for relaxation, capabilities and clinical needs unique to each participant, as assessed by the MT during the first two sessions. 

New, modified for best tolerance, versions of the exercise sets were provided to the participants before the end of the treatment period, at the eleventh session, to support further, independent practice. The contact details of local music therapists experienced with ALS were provided to help facilitate independent practice post-study period.

### 2.12. Equipment, Transportation and Setup

Equipment necessary for the MT treatment include an acoustic guitar, a melodica, a smartphone (with a music player and a metronome application), a pocket-sized speaker for sound amplification and a binder containing visual aids for the anatomy and physiology of singing, the treatment protocol, assessment scales and data entry forms. The equipment are lightweight, easily transportable by urban public transportation and require minimal space and time for setup.

Standard portable equipment, including spirometers and data entry forms, were used by qualified nurses to collect biometric data during regular home visits. Participants travelled to a laboratory for videofluoroscopic swallowing study (VFSS). Voice samples were recorded with a headset microphone, a tablet computer and an audio interface in the patient’s home. Structured interviews were typed by a trained research assistant on a tablet computer.

### 2.13. Feasibility Data Collection and Analysis

Quantitative and qualitative data were collected on recruitment, retention and adherence. Target recruitment over 80%, a retention rate over 70% at the end of the follow-up period and attendance at over 75% of MT sessions delivered were considered the markers of a successful feasibility trial [96,97,98,99]. Self-reported adherence to a suggested, but not required, independent exercise routine was recorded to assess levels of self-motivation that participants demonstrated with regard to music therapy treatment. Any attempt to practice was recorded, even if the exercise set was not fully completed. No fidelity data were collected. No strategies to increase or to control adherence were employed. 

The short-term tolerability of the treatment protocol was assessed by measuring changes in ratings on the Ease of Respiration Numerical Rating Scale and the Ease of Speech Numerical Rating Scale (Appendix D. Numerical Rating Scale for Current Perceived Ease of Respiration (English Translation) and Appendix E. Numerical Rating Scale for Current Perceived Ease of Speech (English Translation)), self-reported before and after every MT session. 

Structured interviews were conducted pre- and post-treatment to assess the treatment experience of participants (pwALS) and the perspective of their caregivers (see Appendix F. Interviews with Research Participants—Persons with ALS, Conducted Prior to Treatment (Week 5) (English Translation), Appendix G. Interviews with Research Participants—Persons with ALS, Conducted at the End of the Follow-Up Period (Week 16) (English Translation), Appendix H. Interviews with Research Participants—Caregivers of Persons with ALS, Conducted Prior to Treatment (Week 5) (English Translation), and Appendix I. Interviews with Research Participants—Caregivers of Persons with ALS, Conducted at the End of the Follow-Up Period (Week 16) (English Translation)). Reflexive thematic analysis (TA) [100] has been employed to analyse the interviews regarding the treatment experience of research participants—pwALS and their caregivers. 

TA is a common method for analysing qualitative data in various research fields. It primarily focuses on identifying and reporting patterns of meaning (themes) within data. A theme captures an important aspect of the data in relation to the research question and presents a patterned meaning within the dataset. The themes may be determined in a number of ways, though the process has to be consistent within each particular TA.

The music therapist’s perspective was assessed through narrative thematic analysis of the individual treatment notes and of the generalised “field notes” submitted by the therapist twice weekly. Narrative TA [101] was applied to the treatment notes and the field notes submitted by the music therapist—principal investigator. Narrative inquiry emphasises preserving the integrity of a particular individual in the course of the analysis. This is essential when researching a novel treatment protocol realised within an ongoing therapeutic relationship. 

### 2.14. Biomedical Data Collection

Assessing bulbar and respiratory dysfunction in ALS presents a challenge for a researcher. Many ALS-specific measurement tools do exist, but, as related research [76,102,103] and problem-oriented discussions in the professional community have revealed (e.g., Bulbar Guidelines Development Symposium, 29th International Symposium on ALS/MND, December 2018), these may not be sensitive enough to reliably measure the change, given that ALS is a rapidly degenerating disease and that the rate of deterioration varies greatly from patient to patient. Predicting the rate of speech and swallow decline has been reported to be especially challenging for clinicians [14,104]. Similarly, although the respiratory subscale of ALSFRS-R is routinely used worldwide to monitor symptoms of respiratory involvement, it only provides limited and sometimes misleading information [105]. Recently published “Provisional best practices guidelines for the evaluation of bulbar dysfunction in Amyotrophic Lateral Sclerosis” [106] present the first uniform assessment of speech and swallow function recommended for multidisciplinary ALS teams.

A minimally burdensome for the participants, inexpensive and reliable assessment battery to measure bulbar and respiratory changes in early- and mid-stage ALS was selected, based on the analysis of existing research, including [5,81,85,87,88,89,107,108,109,110,111,112,113,114,115,116,117,118,119,120,121,122,123]. Taken into consideration were the availability of technical means and local laboratory capacities, as well as local clinical assessment standards, as, for example, nurses being accustomed to routinely measuring Forced Vital Capacity rather than Slow Vital Capacity. For ethical considerations, home-based assessments were preferred whenever possible, with the exception of a VFSS, a dynamic X-ray that allows visualisation of bolus flow and swallowing physiology, for the evaluation of swallowing safety and efficiency [106]. 

Raw data were pseudonymised, protected by password and securely stored in a computer system at ALS Centre Moscow, in accordance with General Data Protection Regulation. A keyword system was developed to pseudonymise the physiological measurements and acoustic samples. The system was designed with the purpose to blind the analysts to the data collection point sequence. It was not possible to blind the data collectors to the condition and treatment of the participants due to the transparent, community-oriented nature of multidisciplinary team care practiced at the host facility.

### 2.15. Outcome Measures to Assess Long-Term Changes in Respiration

Maximal Inspiratory Pressure (MIP), Maximal Expiratory Pressure (MEP), and Forced Vital Capacity (FVC) were measured at four time points to assess long-term changes in respiration. 

### 2.16. Outcome Measures to Assess Long-Term Changes in Cough

Peak Expiratory Flow (PEF) was measured at four time points to assess long-term changes in cough. 

### 2.17. Outcome Measures to Assess Long-Term Changes in Swallowing

The Center for Neurologic Study Bulbar Function Scale (CNS-BFS) swallowing subscore, recorded at four time points, and VFSS, conducted at three time points, were used to assess long-term changes in swallowing. VFSS was conducted by a qualified speech and language specialist in a laboratory in Moscow, using a BV Pulsera Mobile C-arm fluoroscope, pulsing at 30 pulses per second and recorded on a built-in Medical DVD Recorder at 30 frames per second. A volume of 10 mL nectar and pudding boluses were prepared with Nestle (TUC-xanthan gum) at 100% *w*/*v* barium concentration. Transportation and trained volunteer assistance was provided for each participant during the visits to a laboratory for VFSS. 

VFSS video clips were reviewed and scored by a trained speech-language pathologist blinded to the sequence, using frame-by-frame analysis following operational definitions outlined by [124]. Pixel-based measures were anatomically referenced, and expressed as a percent relative to the height of cervical vertebrae C2 to C4 (i.e., %C2–4). The following measures were obtained: Change in Time-to-Laryngeal Vestibule Closure (ms), Change in Maximum Pharyngeal Constriction Area (%C2–42), Change in Peak Position of the Hyoid Bone (%C2–4), Change in Penetration–Aspiration Scale Score (worst) [125], Change in Total Pharyngeal Residue C24 area, measured in percent (%C2–4).

### 2.18. Outcome Measures to Assess Long-Term Changes in Speech

The Center for Neurologic Study Bulbar Function Scale (CNS-BFS) speech subscore, recorded at four time points, and voice samples, recorded at four time points as high-definition .wav files, were used to assess long-term changes in speech. The following voice samples were recorded: spontaneous speech (2 min), passage reading (2 min), /pa/, /ta/, /ka/ syllables sequence repeated as clearly, as fast, as many times as possible on one exhalation (two attempts, the best is used for calculations), /ba/ syllable repeated as clear, as fast, as many times as possible on one exhalation (2 attempts, the best is used for calculations), /a/ sound sustained for as long as possible, at participant’s most comfortable pitch level, on one exhalation (two attempts, the best is used for calculations), separate vowels (A, E, I, O, U) uttered in sequence, with pauses in between, as clearly as possible (two attempts, the best is used for calculations). Since Russian was the native language for all the participants, The Phonetically Representative Russian Text For Fundamental and Applied Studies of Russian Speech created by [126] was substituted for Rainbow passage or Bamboo passage routinely used in speech-language therapy practice in the UK and in the USA (see Appendix J. Text for Oral Reading Task (Russian) for the full text). 

Voice samples were recorded digitally in .wav format, using a Shure WH20XLR Dynamic Headset Microphone, Alesis IO Dock audio interface, Apple iPad 2 tablet and GarageBand software. Participants were encouraged to rest and hydrate between the various recording tasks in order to avoid fatigue and vocal strain. This setup allows control of the sound intensity levels and the distance between the participant’s mouth and the microphone head [70], thus ensuring high quality and consistency of the recordings. 

Acoustic analysis of the voice samples was conducted using the PRAAT linguistic computer program—a scientific tool for analysing speech spectrograms [127]. Compared to perceptual analysis, acoustic speech analysis of sound waveform offers the advantage of describing the voice objectively [128,129,130]. Acoustic analysis of the recorded voice samples in the PRAAT computer program was performed to calculate: Change in Maximum Phonation Time (MPT), sound /a/, measured in seconds; Change in Maximum Repetition Rate—Alternating (MRR-A), /pataka/ sequence, measured in total number of syllables uttered; Change in Maximum Repetition Rate—Sequential (MRR-S), /ba/ syllable, measured in total number of syllables uttered; Change in Fundamental frequency (F0), measured in Hz; Change in Jitter, local, measured in percent; Change in Shimmer, local, measured in percent; Change in Harmonics-to-Noise Ratio (HNR), measured in Db; Change in Vowel Space Area, measured squared Hz, Change in Speaking rate, measured in words per minute; Change in Articulation rate, measured in words per minute; Change in Speech–pause ratio, measured in milliseconds per minute; Change in Pause frequency, measured in number of pauses per minute. 

Perceptual analysis of the same voice samples was conducted by three qualified speech pathologists to assess Change in hypernasality level, measured in points on the scale from “1” (severe hypernasality) to “4” (normal resonance), and the inter-rater reliability was calculated.

For a detailed description of the outcome measures, see Table 2. Outcome measures and data collection summary table.

### 2.19. Biomedical Data Analysis

Extensive secondary (biomedical) data to evaluate respiratory and bulbar changes were collected at multiple time points to provide a perspective on data collection in a home-based clinical trial. This process allowed assessment of the choice of measurement outcomes and provided a perspective on data collection. Only descriptive statistics were used to present and analyse the biomedical data, since, due to the small number of participants (*n* = 7) and the heterogeneity of the group, any statistical inference would be unreliable.

All the biomedical measurements for this study, with the exception of the videofluoroscopic swallowing study, were obtained at four time points:Time point 1 = baseline (week 1),Time point 2 = pre-treatment (week 6),Time point 3 = post-treatment (week 12), andTime point 4 = at the end of the follow-up period (week 16).

The three periods between the time points were marked as the “run-in” (weeks 1–6), “treatment” (weeks 7–12), and “follow-up” (weeks 13–16) periods. 

A laboratory-based videofluoroscopic swallowing study (VFSS) was conducted only at three time points, in order to minimise the burden for the participants, as this test required transportation to the lab. VFSS measurements were obtained at three time points:Time point 1 = baseline (week 1), Time point 2 = pre-treatment (week 6), andTime point 3 = post-treatment (week 12). 

The two periods between these time points are the same “run-in” (weeks 1–6) and “treatment” (weeks 7–12) periods, with the “follow-up” period omitted. 

Changes in trends rather than the numerical values were used to assess the effectiveness of the treatment, since functional decline is expected with ALS progression. When interpreting the long-term data outcomes, it was understood that the direction and the rate of change observed during the run-in period is expected to stay similar if MT treatment has no effect. If MT had been beneficial for a participant, the trend would have been reversed. If the trend was unchanged or amplified, the reason could be that MT had had a negative effect or that participants’ condition had declined rapidly. 

To assess the change in trends, for each of three (or two, in case of videofluoroscopic swallowing study) periods a trend was calculated as a rate of change of the measurement over time (in weeks):Trend_run-in_ = (M_t2_ − M_t1_)/6, Trend_treatment_ = (M_t3_ − M_t2_)/6, andTrend_follow-up_ = (M_t4_ − M_t3_)/4. where M_tn_ is the measurement value at time point n. The lengths of the “run-in” and “treatment” periods are six weeks and the “follow-up” period is four weeks, hence the difference denominators in the formulas.

## 3. Main Study Findings

### 3.1. Demographics

Six females and two males were recruited. Seven presented with spinal ALS onset type, and one participant presented with bulbar onset. The mean age of the participants was 58.1 years. Total ALSFRS-R scores of the participants ranged from 31 to 42 points at recruitment, representing the range of motor and respiratory abilities of the participants. Seven participants completed this study.

Six caregivers were recruited. The relations of the participating caregivers to the participants—persons with ALS—were as follows: four spouses (two wives and two husbands) and two adult children (daughters in both cases). Five caregivers lived in the same household (city apartment) with their relative who had ALS.

Only the data from the seven participants who completed this study were analysed.

### 3.2. Feasibility Data Analysis Results

Analysis of the primary data provides the evidence that it is feasible to implement the suggested study protocol for home-based music therapy as an intervention to support respiratory and bulbar functions in early- and mid-stage ALS. The recruitment rate was 100%, the retention rate was 87.5% and mean adherence was 95.4%—values above the target markers for the protocol feasibility.

Data on the short-term tolerability of the treatment protocol indicated that the protocol was generally well tolerated by the study participants, with the overall trends pointing upwards for both the Ease of Respiration Numerical Rating Scale (*p*-value = 0.296) and the Ease of Speech Numerical Rating Scale (*p*-value = 0.270), even though, as expected in a small study with only seven subjects, neither change was statistically significant. Mean adherence to suggested independent exercise routine constituted 53%. It is indicative that all the participants attempted the independent exercise routine. Common reasons for not engaging into the independent practice routine included not feeling well (e.g., due to high blood pressure and menses), being too busy to exercise (due to work, health care or other commitments), and being too tired to exercise. No participants mentioned the provided individual set of exercises being too challenging among the reasons for not engaging into independent practice routine.

Following reflexive TA, all the participants reported MT to be a pleasurable experience and perceived the suggested independent music therapy exercises as easy to perform. Most pwALS believed that MT was beneficial for their speech and respiration and that it helped them learn new breathing and vocal skills, but did not affect their swallowing. Some participants perceived that MT improved their communication. Most caregivers reported that MT improved or sustained bulbar and respiratory functions of the participants with ALS. Some caregivers reported that the MT process and, in particular, communication with the music therapist had a positive effect on the psychological state of the pwALS, and that participation of their family members affected by ALS in MT had a positive effect on caregiver’s own psychological state.

Brief narrative accounts of individual music therapy sessions were submitted by the principal researcher after every treatment session, total of 80. Additionally, field notes—generalised reflections on the course of research and tendencies within the group during the treatment phase—were submitted by the principal researcher, total of 13. The following themes were analysed in detail: (1) logistics of the suggested home-based MT treatment, (2) essential professional self-care of music therapist, (3) experimental treatment protocol implementation (including synchronous progress in the cohort, music preferences and song choice, individual treatment protocol adjustments, and audio and video recording of the sessions), (4) the music-assisted relaxation process, (5) post-treatment exercises for individual practice, and (6) the role of music therapist in the protocol implementation. 

The results of the narrative TA indicate that the MT protocol is safe, if implemented with necessary individual modifications by a professional music therapist. 

### 3.3. Biomedical Data Analysis Results

Most bulbar and respiratory functional parameters were sustained or improved at a higher rate during the treatment period, as compared to the run-in (control) period, as evident from the mean trends for the 32 biomedical measurement parameters across respiratory, speech and swallowing domains.

### 3.4. Long-Term Changes in Respiration

FVC, MIP, and MEP were measured at the four time points to assess long-term changes in respiration. Mean trends for all three outcome measures suggest that respiratory functional parameters of the study participants were sustained or improved during the treatment period. For a summary of the individual and mean measurements at four assessment points, see Table 3 for FVC (%), Table 4 for MIP (cm H_2_O) and Table 5 for MEP (cm H_2_O).

### 3.5. Long-Term Changes in Cough

Peak Expiratory Flow (PEF) was measured at the four time points to assess long-term changes in cough. Mean trends suggest that the cough functional parameters of the study participants improved during the treatment period. For a summary of the individual and mean measurements at four assessment points, see Table 6. PEF (%) individual and mean measurements at four assessment time points.

### 3.6. Long-Term Changes in Speech

The Center for Neurologic Study Bulbar Function Scale (CNS-BFS) speech subscore was measured at four time points to assess long-term changes in speech. A lower score corresponds to better functioning. For a summary of the individual and mean measurements at four assessment points, see Table 7. CNS-BFS speech subscore (points) individual and mean measurements at four assessment time points.

Voice samples were recorded at four time points to assess long-term changes in speech and resulted in the following outcomes:(1)Maximum Phonation Time (MPT), sound /a/, measured in seconds, higher score corresponds to better functioning;(2)Jitter, local, sound /a/, measured in percent, lower score corresponds to better functioning;(3)Shimmer, local, sound /a/, measured in percent, lower score corresponds to better functioning;(4)Harmonics-to-Noise Ratio (HNR), sound /a/ measured in Db, higher score corresponds to better functioning;(5)Maximum Repetition Rate—Alternating (AMR), /pataka/ sequence, measured in total number of syllables uttered, higher score corresponds to better functioning;(6)Maximum Repetition Rate—Sequential (SMR), /ba/ syllable, measured in total number of syllables uttered, higher score corresponds to better functioning;(7)Vowel Space Area, vowels /a, e, i, o, u/, measured in squared Hz, higher score corresponds to better functioning;(8)Fundamental frequency (F0), oral reading, measured in Hz, higher score corresponds to better functioning;(9)Speaking rate, oral reading, measured in words per minute, higher score corresponds to better functioning;(10)Speech–pause ratio, oral reading, measured in seconds per minute, lower score corresponds to better functioning;(11)Pause frequency, oral reading, measured in number of pauses per minute, lower score corresponds to better functioning;(12)Hypernasality level, oral reading, measured in points, higher score corresponds to better functioning.

Initially, two minutes of oral passage reading and two minutes of spontaneous speech were included in the list of recorded voice samples. Although these tasks presented no issues for most participants, some were fatigued by the two-minute oral reading, gasping for breath and reading in a quieter or strained voice by the end of the first minute. Others became too emotional when recording spontaneous speech, changing the subject very soon from the given neutral theme to a discussion of their disease progression or family issues. For ethical considerations, the decision was made not to record spontaneous speech and to shorten the oral reading passage to include only the first five sentences. All participants needed more than 30 s to complete reading this passage. 

Perceptual analysis of the voice samples was conducted by three qualified speech pathologists, native Russian speakers, to assess changes in hypernasality levels, measured in points on the scale from “1” (severe hypernasality) to “4” (normal resonance). The inter-rater reliability between the three raters was calculated and was low at 52%.

For a summary of the individual and mean speech measurement outcomes at four assessment points, see Table 8 for MPT (seconds), Table 9 for Jitter (local, %), Table 10 for Shimmer (local, %), Table 11 for HNR (Db), Table 12 for MRR-A (syllables, total), Table 13 for MRR-S (syllables, total), Table 14 for VSA (Hertz, squared), Table 15 for Fundamental frequency (Hz), Table 16 for Speaking rate (words per minute), Table 17 for Speech–pause ratio (seconds per minute), and Table 18 for hypernasality level (points).

### 3.7. Long-Term Changes in Swallowing

The Center for Neurologic Study Bulbar Function Scale (CNS-BFS) swallowing subscore, recorded at four time points, was used to assess long-term changes in swallowing. See Table 19 for individual and mean scores. A lower score corresponds to better functioning. 

Videofluoroscopic swallowing study (VFSS) video clips were recorded at three time points to assess long-term changes in swallowing and resulted in the following outcomes:(1)Time-to-Laryngeal Vestibule Closure, nectar 10 mL, measured in ms, lower score corresponds to better functioning;(2)Time-to-Laryngeal Vestibule Closure, pudding 10 mL, measured in ms, lower score corresponds to better functioning;(3)Maximum Pharyngeal Constriction Area, nectar 10 mL, measured in %C2–4^2^ lower score corresponds to better functioning;(4)Maximum Pharyngeal Constriction Area, pudding 10 mL, measured in %C2–4^2^, lower score corresponds to better functioning;(5)Peak Position of the Hyoid Bone, nectar 10 mL, measured in %C2–4 higher score corresponds to better functioning;(6)Peak Position of the Hyoid Bone, pudding 10 mL, measured in %C2–4, higher score corresponds to better functioning;(7)Penetration–Aspiration Scale Score (worst), nectar 10 mL, measured in points lower score corresponds to better functioning;(8)Penetration–Aspiration Scale Score (worst), pudding 10 mL, measured in points, lower score corresponds to better functioning;(9)Total Pharyngeal Residue C24area, nectar 10 mL, measured in %C2–4, lower score corresponds to better functioning;(10)Total Pharyngeal Residue C24area, pudding 10 mL, measured in %C2–4, lower score corresponds to better functioning;(11)Laryngeal vestibule closure, nectar 10 mL, described as complete, partial, or incomplete, LVC described as “complete” corresponds to a safe swallow;(12)Laryngeal vestibule closure, pudding 10 mL, described as complete, partial, or incomplete, LVC described as “complete” corresponds to a safe swallow.

All participants who completed this study made all three visits to the laboratory for VFSS and followed the data collection protocol. However, consistent data from all three time points are only available for five participants, for two samples (nectar and pudding). The data are missing due to a mistake in video capturing: some clips were only recorded by an external video camera, but not by the built-in Medical DVD recorded of the BV Pulsera Mobile C-arm fluoroscope. The quality of the video from the external camera was sufficient for descriptive clinical swallowing assessment by a speech and language specialist, but too poor for the calculations used in this study to take place. For the participants with the missing high-quality video recordings, no VFSS data were analysed. No VFSS data were analysed for liquid samples. 

For a summary of the individual and mean swallowing measurement outcomes at three assessment points, see Table 20 for LVCrt, nectar 10 mL (ms); Table 21 for LVCrt pudding 10 mL (ms); Table 22 for MPCAn, nectar 10 mL (%C2–42); Table 23 for MPCAn, pudding 10 mL (%C2–42); Table 24 for PeakXY, nectar 10 mL (%C2–4); Table 25 for PeakXY, pudding 10 mL (%C2–4); Table 26 for PAS (worst), nectar 10 mL (points); Table 27 for PAS (worst), pudding 10 mL (points); Table 28 for TotC24, nectar 10 mL (%C2–4); Table 29 for TotC24, pudding 10 mL (%C2–4); Table 30 for LVC, nectar 10 mL; Table 31 for LVC, pudding 10 mL.

## 4. Discussion

### 4.1. The Feasibility of the Suggested Study Protocol

Primary data analysis provides first evidence that it is feasible to use the suggested study protocol for home-based MT as an intervention to support respiratory and bulbar function in early- and mid-stage ALS.

An innovative, ALS-specific MT protocol was developed by the principal researcher and used as a treatment intervention in this study. A MT session following the suggested protocol resembles an adaptive voice lesson, where a music tutor works with a singer to improve posture, breath support, resonance, intonation, articulation and voice projection. However, this protocol, through the use of music, musical structures and musical interactions, focuses on non-musical, rehabilitative goals and objectives. These goals, which are ALS specific and common for the whole cohort, are to increase breath support and muscle relaxation, to maintain speech rate, prevent or decrease hypernasality, and to maintain swallow coordination. The individual objectives were related to these goals, but depended on the particular symptoms, progress, actual assessment and the observed current state of each participant. Data on the short-term tolerability of the treatment protocol indicate that the protocol was generally well tolerated. Additionally, all the participants reported MT to be a pleasurable experience and perceived the suggested independent MT exercises as easy to perform. This is consistent with previous findings that musical cueing during physical exercises decreased perceived exertion and perceived fatigue, as compared to treatment not supported by music [131,132].

### 4.2. Unique Properties of MT Interventions and the Potential Role of MT in MDT ALS Care 

Facilitating musical structures, mostly in the form of the guitar accompaniment, make this MT treatment inherently different from the speech therapy or the physical therapy process. Priming the ability of an external auditory cue to stimulate recruitment of motor neurons at the spinal cord level, and cueing of the movement period by rhythmic synchronisation throughout the whole duration and trajectory of the movement [133,134] are the essential mechanisms behind the use of music in sensorimotor rehabilitation. Rhythmic stimulation increases endurance and motivation [135], and enhances memory and cognition [136]. Rhythm can be employed to improve temporal characteristics of speaking, such as fluency, articulatory rate, pause time and intelligibility of speaking [53]. Whilst rhythm is essential, there are many more parameter characteristics to the facilitating musical structures. These include the tempo, pitch, tonality and rhythm changes and can be modified by the music therapist during the session in various ways to best support the work on the rehabilitative objectives for each participant. For example, the range of a singing exercise can be expanded after the participant has learned to better control their laryngeal elevation or the tempo in melodica exercise could be decreased to allow longer phrases, which means longer exhalation (i.e., potentially increased lung capacity). Additionally, vice versa, the tempo of singing exercises can be gradually reduced to accommodate the deterioration of function experienced as a result of disease progression, still allowing the work on articulation. One example of the unique effect music-supported therapy has is the fact that for all the participants, their comfortable singing range was more extensive than their speaking range. It appears to be safe and effective to ask participants to sing slightly higher than their normal (as affected by ALS) speaking range.

Unlike a vocal coach, the therapist has to be similarly attentive to the non-musical elements of the protocol. For example, if a participant presents with severe hypernasality, the music therapist can replace the “hah-mah-mah” sequence with the “hah-gah-gah” sequence in a singing exercise to help the participant strengthen their velopharyngeal closure; or, if excessive jaw muscle tightness is a problem, the participant can be instructed to support the jaw with their hand. To make these modifications, the therapist has to be a skilled clinician and to have a deep theoretical understanding of the anatomical and neurological mechanisms behind the bulbar and respiratory functional changes in ALS.

Proficiency in MT requires the complex combination of knowledge that includes music perception and production mechanisms, music composition, performance and goal-oriented improvisation, basic psychology, psychopathology, social science, anatomy, physiology and neurology, treatment planning, documentation, research and ethics. This diverse expertise allows music therapists to plan music interventions and to provide and adjust musical structures in real time to support health goals in the context of multidisciplinary ALS care.

The therapeutic relationship between a therapist and a client is always at the core of the MT process, however structured and prescriptive the therapist-led protocol may be. As follows from the TA of the individual treatment notes and generalised “field notes” submitted by the music therapist, most participants’ need for authentic communication, validation and open exploration of challenging personal ALS-related topics appeared immense. Only one of the participants enjoyed strong psychological support available from the family. Keeping the balance of providing basic psycho-emotional support enough for each participant to work on their therapy goals without switching over to music psychotherapy presents a professional objective for a music therapist when implementing this protocol. Additionally, it highlights the need for formal psychological support services being readily available for pwALS.

### 4.3. Study Limitations

The small sample size is an obvious limitation of this feasibility study.

There are certain limitations to this research resulting from the disparity of treatment and evaluation between ALS clinical facilities around the world. In particular, this resulted in missing VFSS data, absence of reliable tools to assess nasalance acoustically and use of the Center for Neurologic Study Bulbar Function Scale (CNS-BFS) which was not validated for use in Russian at the time this study was conducted.

With the exception of VFSS data, data analysis for this study was ultimately performed by the principal researcher, who also developed the experimental protocol and provided the treatment. Combining these roles could lead to biased interpretation of the data. 

The interviews were conducted in Russian, the language native to the participants and the research assistants, and translated into English by the principal researcher, who speaks both Russian and English. This could result in language and cultural bias.

Finally, the duration of the follow-up phase was planned to be shorter (at four weeks) than the run-in and the treatment phases (six weeks each), due to the cultural context. In the summer, most people in Russia who are unemployed move temporarily to their country houses. Extending the follow-up period to six weeks could prevent follow-up data collection.

### 4.4. Measuring Changes in Bulbar Functions in ALS

The biomedical data were collected as secondary data in this study, in order to understand if the chosen outcome measurements and parameters were feasible and sensitive enough to measure changes in bulbar and respiratory functioning in ALS across the 16 weeks. Whilst respiratory function assessment tools for ALS are largely standardised, evaluation of bulbar dysfunction in ALS remains the central topic of ALS clinical care and research.

Voice samples were used to assess long-term changes in speech for this study. Acoustic analysis of voice samples in the PRAAT computer program was preferred to clinician-based perceptual analysis for objectivity and logistic convenience [128,129,130]. Whilst instrumental evaluations, including acoustic, aerodynamic and kinematic methodologies [86,122] can provide even more precise information regarding bulbar decline in ALS, these require a trained specialist’s presence, may be invasive, and require the purchase of specialist and expensive equipment and materials. Voice sampling requires less expensive and less specialised equipment. The process of recording takes less than 10 min of participants’ time and does not require the presence of a highly trained speech-language specialist at the point of data collection. Acoustic analysis of these voice samples has to be planned and carried out by a qualified phonetician or speech-language specialist who is blinded to the time order of the recordings. Such analysis is a largely automated, objective, fast and inexpensive way to assess speech in ALS when instrumental evaluation is not possible. One important exception to this is nasalance assessment. Increased nasalance (hypernasality) is a prominent characteristic of speech decline in ALS and directly affects speech intelligibility [137]. Acoustic analysis does not offer consistent assessment of hypernasality in speech, unlike other speech parameters. Clinician-based analysis of the recorded samples was conducted for this study to assess nasalance of the voice samples perceptually. Low inter-rater reliability was calculated between the three raters, who were qualified speech pathologists. It may be concluded that perceptual analysis of nasalance in ALS is not reliable as well. Aerodynamic measure of nasal flow has been recently reported to be the most efficient way to evaluate velopharyngeal dysfunction in ALS [138] and can be recommended for evaluation of nasalance in clinical practice and in future research.

VFSS has been recommended in the guidelines document as the most accurate tool to assess the risk for silent aspiration and dysphagia in people with ALS [104]. This is a laboratory-based test, which may impose more burden onto the participants than home-based tests. The participants are required to travel to a medical facility, the procedure takes at least 30 min, and some participants may find the taste and consistency of the barium unpleasant. However, our study demonstrates that, with adequate support from a community-based ALS care centre, the VFSS procedure was well tolerated by ALS patients. Such support included volunteer assistance with transportation and navigating the non-accessible urban environment. All the participants chose to take part in all three VFSS assessments during the first 12 weeks of this study. Frame-by-frame analysis of video clips recorded during VFSS was used to assess long-term changes in swallowing for this study. The analysis was performed during the data analysis phase of this study by a highly qualified speech-language pathologist who was blinded to the time order of the video clip recordings and used published operational definitions [124]. Such specialist analysis of VFSS video clips has been reported to be a reliable [139] and feasible way to evaluate swallowing dysfunction in ALS. 

To summarise, the standard home-based instrumental respiratory tests, the videofluoroscopic study (VFSS) with subsequent frame-by-frame analysis of video clips, the acoustic analysis of recorded voice samples, along with self-reported scales such as CNS-BFS used in this study appear to be a feasible and effective way to assess respiratory and bulbar function in ALS. Aerodynamic measure of nasal flow can be recommended for reliable evaluation of nasalance in ALS.

### 4.5. The Need for an Efficacy Study

Although most mean trends in biomedical data suggest that the bulbar and respiratory functions of the group participants were sustained or improved during the treatment period, whilst, in most cases, the same functions deteriorated during the run-in and follow-up periods, no general assumptions should be drawn. The sample size of this feasibility study is very small, especially considering the partially missing VFSS data, and the individual responses of the participants to treatment vary. A study with the main goal of understanding the effect of the described MT protocol on bulbar and respiratory dysfunction in ALS has to be designed with a larger cohort.

Whilst randomised control trials are considered the gold standard for evidence-based practice, there are limitations to this design specific to music therapy treatment for ALS. For example, blinding of the participants is not possible in a music therapy trial [140] and there are ethical implications for randomisation of the palliative patients into the control group, since no other comprehensive treatment for bulbar and respiratory decline in ALS was found in the literature. ALS presentation and progression are very heterogeneous and existing measures of assessment of ALS progression have limited sensitivity, which results in the need for long trials with large sample sizes. A better understanding of the disease would improve the overall clinical study design and allow for small sample sizes, for example, through stratification based on predictive baseline values [72]. Every person with ALS follows their own pattern of disease progression, and it is the individual rate of decline to date that may, to some extent, predict their future decline. It can be argued therefore that the design of the subsequent study should employ the same ABA design as the current study, where the participants serve as their own controls. 

### 4.6. Suggested Study Protocol Modifications

Based on the feasibility data from the current study, it is strongly advised that a mental health assessment is added to the list of the inclusion criteria for study of efficacy. Screening for depression, anxiety, and hopelessness, and availability of social support and psychiatric illness at recruitment need to be considered [141].

It can be recommended that no spontaneous speech samples are collected for the subsequent study and that the oral reading task is limited to a one-minute sample. This is an important ethical consideration, with the goal to prevent voice fatigue and potential emotional vulnerability of the participants, as discussed above. The Bamboo passage can be recommended to be used as the oral reading task for native English speakers [142]. Analogous passages will have to be identified to evaluate the change in speech parameters in other languages.

Table 32 presents the suggested list of outcome measures and parameters to assess the effect of the MT treatment protocol on the bulbar and respiratory functions of the participants for the efficacy study. For all the outcome measures, the data have to be collected at four points: week 1 (baseline), week 6 (beginning of treatment), week 12 (end of treatment) and week 18 (end of washout).

## 5. Conclusions

This is the first biomedical MT study involving pwALS and the first study of any kind to systematically look at supporting bulbar and respiratory functions in ALS. The data from the 16 week ABA design feasibility trial suggest that the MT protocol has no adverse effect on bulbar and respiratory functions of persons with ALS, in the short term or in the long term, and that the home-based treatment was perceived as pleasant, motivating and beneficial by the seven research participants. A comprehensive battery of outcome measurements and parameters to reliably evaluate the therapy-induced change in bulbar and respiratory functions was defined. Using these measurements, the biomedical data were collected as secondary data in this trial. Positive trends for bulbar and respiratory parameters were observed as a result of the MT treatment. The study protocol was feasible, with minor modifications. A subsequent large-cohort study with the aim of assessing the effectiveness of the treatment protocol is warranted.

## Figures and Tables

**Table 1 brainsci-12-00494-t001:** Experimental music therapy treatment protocol to support bulbar and respiratory functions of persons with early- and mid-stage amyotrophic lateral sclerosis.

**I. Session Opening and Assessment**
**Time** (approximately): 5 min.**Therapy objective**: Assessment.**Materials and equipment**: NRS (respiration) and NRS (voice) sheets; NRS data sheet; notebook for note taking. Two sturdy chairs situated facing each other, approximately 1.5 m apart. Optional: wheelchair, small table, pillow for participant’s comfort.**Procedure**: Music therapist (MT) and participant exchange salutations. MT assesses participant’s physical and emotional state through observation and conversation, and re-establishes rapport through a brief conversation to ensure psychological comfort of the participant entering the session. The following information is recorded: (1) NRS for current perceived ease of respiration, (2) NRS for current perceived ease of voice production, (3) information about adherence to the assigned independent exercises routine: frequency, duration, difficulties, comments (starting at session 4). Participant is reminded that he/she is going to be guided through all the exercises and is welcome to participate to his/her comfort, to ask clarifying questions and make comments, to stop doing an exercise at any point if he/she feels uncomfortable or tired, and to pause, rest and hydrate as needed.
**II. Body Alignment Exercise**
**Time** (approximately): 3 min.**Therapy objective**: To learn proper body alignment and its role in respiration, voice production and swallowing.**Materials and equipment**: Visual aids for anatomy of singing.**Procedure**: II.1. Body awareness. Participant is encouraged to become aware of the physical sensation of his/her body, to pay close attention to any muscle tension, strain or stiffness and to gently move, stretch or self-massage to release those. II.2. Body alignment Participant is encouraged to become aware of his/her body alignment and to find a good sitting posture by maintaining the spinal alignment from the hips up. Suggested steps to achieve this are: a. Both feet are on the ground, shoulder-width apart, forming a 90-degree angle with the thighs; b. The pelvis is adjusted so there is slightly less curve in the lower back, and the spine feels extended both upwards and downwards; c. The rib cage is now more upward and “open” (not collapsed); d. The shoulders are suspended exactly over the rib cage (rather than pulled back or rolled forward); e. The head is balanced at the top of the spine and feels almost weightless, with the front half of the skull is balanced in front of atlanto-occipital joint, and its hind half is balanced behind atlanto-occipital joint; f. The upper body is poised and aligned, yet flexible and ready to move: the images of the whole body as a marionette suspended by a rope extending from the top of the skull, or of the head as a water lily flower resting on water surface may be helpful; g. To check for proper spinal alignment: stretch arms above the head and bend them down so that fingertips of one hand touch the elbow of the opposite arm; sustain this position for several seconds, then let the arms drop down gently to the sides of the body, but keep the posture. The anatomy and physiology of respiration, voice production and swallowing are briefly explained to the patient during these and following exercises in order to increase the patient’s awareness and sense of control over these processes.
**III. Diaphragmatic Breathing Exercises**
**Time** (approximately): 4 min.**Therapy objective**: To become aware of diaphragmatic action, its role in respiration and benefits of diaphragmatic breathing.**Materials and equipment**: Visual aids for anatomy of singing.**Procedure**: III.1. Silent long diaphragmatic breathing (5–10 repetitions) Participant is instructed to breathe in through the nose, with the mouth slightly open, then breathe out through the mouth, without sound. Air enters and escapes from the lungs with no effort: it is not forced in any way, and no rib action should occur. Participant is encouraged to place one hand on his/her abdomen and the other on his/her chest as he/she practices diaphragmatic breathing. The front and the side of the abdominal wall expand as the diaphragm contracts and pulls down during breath in, while no visible movement of the chest or shoulders occurs. III.2. Audible diaphragmatic breathing on [s] sound (3 repetitions) Participant is instructed to breathe in through the nose, with the mouth slightly open, then breathe out through the mouth making a continuous [s]sound until he/she runs out of air. The sound should not be forced, and no rib action should occur. III.3. Silent long diaphragmatic breathing (3 repetitions)—see III.1. III.4. Diaphragmatic breathing with audible sigh (3 repetitions) Participant is instructed to breathe in through the nose, with the mouth slightly open, then breathe out through the mouth making an audible, very breathy sigh on a vowel sound (for example, ‘a’). The sound should not be forced, the throat should be relaxed, and no rib action should occur.Note: No background music or music accompaniment will be used, as this will allow the participants to fully concentrate on the physical sensation of diaphragmatic breathing. During the first several sessions, the participant may experience a slightly uncomfortable pulling sensation around the posterior abdomen wall: this sensation is due to this group of muscles being more intensively worked than usual and will subside. When it becomes habitual, diaphragmatic breathing may reduce the effort necessary for breathing.
**IV. Controlled Breathing and Lip Seal Exercise**
**Time** (approximately): 3 min.**Therapy objective**: To practice controlled breathing in order to create sustained airflow necessary for speech; to improve oxygen and carbon dioxide exchange; to maintain lip seal (necessary for swallowing and for decreasing salivation).**Materials and equipment**: 37-key melodica; individual tube mouthpiece; metronome (Android smartphone, Metronome Beats application for Android, Kinivo ZX120 Mini Portable Wired Speaker for amplification). Optional: a small (approx. 15 cm × 15 cm) piece of lightweight fabric.**Procedure**: IV.1. Pursed lip breathing (3 repetitions) Participant is instructed to take a diaphragmatic breath in through the nose and to breathe out very slowly through pursed lips (“as if blowing on a fire”). A piece of lightweight fabric held by MT in front of the participant can be used for visual feedback. IV.2. Controlled breathing and lip seal exercise Participant is instructed to blow into the tube mouthpiece of the melodica while MT plays on the keys of the melodica the first 16 measures from “Old French Song” from “Children’s Album” (Op.39, No.16) by Tchaikovsky, at 60 bpm, with accompanying audible metronome click. Participant is advised to breathe in through the nose as necessary between the phrases, MT may provide conducting cues for inhale as necessary. The tempo is adjusted as needed (decreased for a longer exhale) to match participants’ ability. 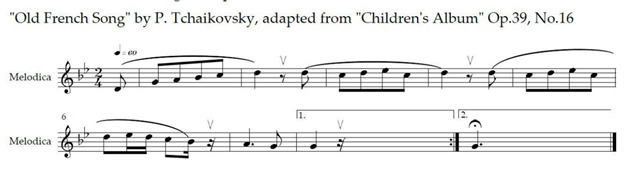
**V. Music-Assisted Relaxation for Voice Production**
**Time** (approximately): 8 min.**Therapy objective**: To elicit the relaxation response through music-assisted visualisation technique (V.1) and to relax and stretch the muscles involved in voice production (V.2–V.8).**Materials and equipment**: Android smartphone; default music player for Android; Spotify application for Android (with subscription); Kinivo ZX120 Mini Portable Wired Speaker for amplification). Optional: lotion or oil for massage; Purell for hand disinfection (if MT touches participant’s face).**Procedure**: Participant is reminded to maintain the aligned sitting posture, and to be gentle rather than forceful in releasing muscle tension.V.1. Music-assisted visualisation and relaxation (about 4 min) The recorded relaxing music is started in the background at the level allowing for a live narrative to be clearly heard. Maintaining the aligned sitting body position, participant is advised to close the eyes and to breathe deeply, as he/she is led by MT through a brief (about 3 min) music-assisted visualisation for relaxation. At the end of the exercise, as the background music continues, participant is invited to gently open his/her eyes, first looking downwards and then gradually orienting him/herself to the surroundings. The background music continues through the rest of the exercise sets. V.2. The head and the neck muscles relaxation (2 repetitions) Maintaining the body alignment, participant is advised to let his/her head slowly fall forward under its weight, to feel the stretch on the spine, and then to bring the head back to the balanced position at the top of the spine. Participant may also sway the head gently side to side during this exercise, if it feels appropriate. V.3. The facial muscles relaxation Participant is advised to rub his/her hands together or to warm them otherwise and to place the palms over closed eyes for several seconds, then to gently massage the face with circular motions of the pads of the fingers, using both hands, moving from hairline downwards to cheeks, lips and chin, spending more time on areas that feel tight. When the cheeks are being massaged, participant is advised to let his/ her jaw to hang slack. Face massage may be performed by MT or a care provider if the patient’s hand function is impaired.V.5. The tongue muscles relaxation and stretch (3 repetitions) Participant is encouraged to let the tongue relax and fall forward slightly out of the mouth by releasing its muscles, then to gently stretch the tongue out of the mouth down toward the chin, and to release again allowing the tongue to rest on the lower lip. Participant is further advised to pull the tongue back into the mouth as far as possible, hold for a few seconds, and release. V.6. The mandible (jaw) relaxation and stretch Participant is assisted in finding his/her temporomandibular joints. With the fingers placed over the joints, the patient allows the jaw to drop without resistance. Participant is encouraged to take his/her jaw between the thumb and forefinger and to gently move the jaw up and down, at first slowly, then faster. The movement will be unrestricted if the jaw is relaxed. Participant is then advised to move the jaw as far forward as possible, hold, then release; then to move the jaw as far back and upwards as possible (chin tuck), hold, then release. This stretch may be repeated 2 or 3 times. V.7. The suprahyoid muscles relaxation Participant is assisted in finding his/her suprahyoid (digastric and mylohyoid) muscles under his/her chin (the muscles responsible for elevating the larynx during swallowing). Participant is advised to gently massage these muscles with his/her thumbs in slow, “kneading” motions, pushing vertically up and releasing down.V.8. The infrahyoid (strap) muscles relaxation Participant is assisted in finding his/her larynx by placing fingers flat against the front of his/ her neck and swallowing, thumb and forefinger of one hand are used to gently move the larynx side to side several times.
**VI. “Ping Pong” Soft Palate Exercise**
**Time** (approximately): 1 min.**Therapy objective**: To tonicise the soft palate muscles involved in velopharyngeal function and to practice the proper soft palate position for phonation. **Materials and equipment**: None.**Procedure**: Participant is instructed to “yawn politely” (half yawn) in order to find the proper position for relaxed sound production [95]. It may be helpful to imagine there is a ping pong ball in the back of the mouth and to hold this position for several seconds. Repeat 5 times.
**VII. Phonation Exercises**
**Time** (approximately): 5 min.**Therapy objective**: To facilitate proper engagement of arytenoid cartilages and vocal folds (VII.1–VII.3), to increase the speech rate (VII.3). **Materials and equipment**: Visual aids for anatomy of singing.**Procedure**: VII.1. “Hah” sigh exercise (2 repetitions) Participant is instructed to take a breath, to expel about half of it, then to add a light, “lazy” sigh (“hah”), starting in the voice midrange and inflecting downwards. The tongue rests in the limp position, the jaw is relaxed. 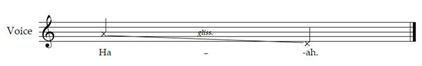 VII.2. Interrupted “hah” sigh exercise (2 repetitions) Participant is instructed to repeat the previous exercise, allowing the lips to close and open several times at the beginning of the sigh, resulting in a light humming sound (“hah-mah-mah-mah-mah”). The lips are not pressed firmly together like in the regular [m] sound. The tongue rests in the limp position, the jaw is relaxed. 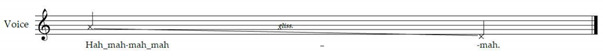 VII.3. Voiced consonant sigh exercise (2 repetitions for each consonant) Maintaining relaxation of facial and laryngeal muscles, participant is instructed to take a diaphragmatic breath, then to expel the air while making a continuous consonant sound, starting in the voice midrange and inflecting downwards. An accompanying short vowel sound has to be added to the stop consonants (‘b’, ‘d’, ‘g’), e.g., “ba-ba-ba-ba-ba”. The sound should not be forced. The jaw and tongue are relaxed. The sequence of the consonants for this exercise is: /v/, /z/, /z</, /l/, rolled /r/, /j/, /b/, /d/, /m/, /n/, /g/. Rolled /r/ may present a challenge for some people. This is not related to ALS and should not discourage the patient from attempting this exercise.
**VIII. Consonant Range Cantillation Exercise**
**Time** (approximately): 2 min.**Therapy objective**: To facilitate tongue movement ease and speed, to improve velopharyngeal function, to prevent (reduce) hypernasality. **Materials and equipment**: None.**Procedure**: Participant is instructed to take a diaphragmatic breath and say “mah-nah-ng-ah” in cantillation once, then proceed saying “mah-nah-ng-ah” 3 times on one exhalation, next—saying it 6 times on one exhalation, then 9 times on one exhalation, and, finally, 12 times on one exhalation. Participant is encouraged to maintain and note the freedom of tongue and jaw movement as he/she does this exercise. Note: The sequence of tongue movements required for this exercise involves fast progression from the resting position (/m/) to hard palate (/n/), to soft palate (/g/) [93]. Besides facilitating tongue movement ease and speed, this exercise has the potential to improve velopharyngeal function and, thus, to prevent (reduce) hypernasality, which also contributes to speech intelligibility [85].
**IX. Velopharyngeal Port Exercise**
**Time** (approximately): 3 min.**Therapy objective**: To improve velopharyngeal function, to prevent hypernasality. **Materials and equipment**: Visual aids for anatomy of singing. Acoustic guitar; metronome (Android smartphone, Metronome Beats application for Android, Kinivo ZX120 Mini Portable Wired Speaker for amplification).**Procedure**: Starting in the lower midrange of his/her voice, participant is instructed to sing the syllables “hun-ga” three times as a scale from sol to do. “Hun” corresponds with the offbeat, and “ga” falls on the beat. The exercise is then modulated gradually up by semitones, until it reaches the limit of the participant’s comfortable range. After that, it is modulated down by semitones until it is three semitones below the starting key. The therapist models the exercise and provides the guitar accompaniment, and sings together with the participant. The patient is encouraged to notice the switching between nasal (“hun”) and non-nasal sound (“ga”). Audible metronome click is setup. The tempo of the accompaniment can be adjusted to the ability of the patient. Gradual increase to up to 90 bpm is advisable in later sessions. 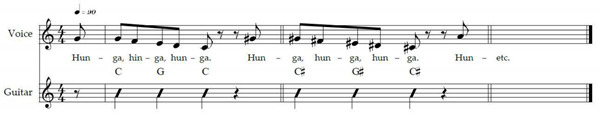
**X. Impulse Diaphragmatic Breathing Exercises**
**Time** (approximately): 2 min.**Therapy objective**: To increase the efficiency and speed of relaxed diaphragmatic inhalation.**Materials and equipment**: Acoustic guitar; metronome (Android smartphone, Metronome Beats application for Android, Kinivo ZX120 Mini Portable Wired Speaker for amplification).**Procedure**: X.1. Impulse breathing with short “hah” exhale (2 repetitions)The exercise starts with participant listening to 8 beats of audible metronome click initially set at 4/4, 64 bpm. Participant is instructed to fully relax abdominal muscles, letting air effortlessly enter the lungs, then to abruptly expel the air with a short, though deep and strong (“barking”) “hah” sound and to immediately let the abdominal muscles relax again, letting air into the lungs. Suggested sequence: (1) listen and rest for 8 beats, (2) 8 ”hah” utterances for 8 beats, (3) listen and rest for 8 beats, (4) 8 ”hah” utterances for 8 beats, (5) stop. The tempo should be slightly faster than the tempo most comfortable for the participant and may be increased gradually in subsequent sessions if the patient is ready. X.2. Impulse breathing with sustained “hah” exhales (4 repetitions)As in the previous exercise, participant is advised to fully relax the abdominal muscles, letting air effortlessly enter the lungs. Then, the air is expelled 3 times following the pattern: “hah-hah-haaaaaaaah” (short-short-long), where /a/ vowel is very open, strong and deep, but is not forced. After each syllable, the abdominal muscles should relax again, letting air into the lungs.
**XI. Sustained Vowels Production Exercises**
**Time** (approximately): 5 min.**Therapy objective**: To practice full diaphragmatic breathing and healthy vocal folds coordination for sustained, supported vowel production.**Materials and equipment**: Acoustic guitar; metronome (Android smartphone, Metronome Beats application for Android, Kinivo ZX120 Mini Portable Wired Speaker for amplification).**Procedure**: XI.1. Vowel shaping exercise (3 repetitions)With the mouth fully closed, participant is instructed to silently form vowel shapes in the following sequence: /a/, /e/, /i/, /o/, /u/, paying attention to position changes in tongue and facial muscles. MT model the vowels (with sound).XI.2. “Hah-meh-mee-moh-moo” sigh exercise (3 repetitions)Similar to exercise VII.3, participant is encouraged to take a breath, then expel it on a light, “lazy” sigh, starting in the voice midrange and inflecting downwards. During this exhale, participant forms the vowel shapes /a/, /e/, /i/, /o/, /u/ in a relaxed manner and allows the lips to close and open several times, resulting in light humming sound: “hah-meh-mee-moh-moo”. The lips are not pressed firmly together like in the regular [m] sound, and the jaw is relaxed. 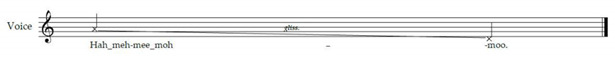 XI.3 “Hah-eh-ee-oh-oo” sigh exercise (3 repetitions) The instructions for this exercise are the same as for the exercise IX.2., but the lip movement resulting in the light /m/ sound is now omitted: “hah-eh-hee-oh-oo”. 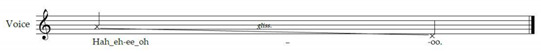 XI.4. Sustained vowels exercise (2 repetitions, if well tolerated) Comfortable tone from participant’s midrange is chosen for the exercise and becomes the tonal center (do). The patient sustains do for four measures (2/4, 90 bpm) on “mah” syllable, then rests for 2 measures. The next syllable (“meh”) is then sustained is a similar manner, then “mee”, “moh” and “moo”. The therapist provides the guitar accompaniment, and sings together with the participant. Accompanying audible metronome is setup at 2/4, 90 bpm. The tempo is adjusted as needed (decreased for a longer exhale) to match participants’ ability. The exercise may be performed twice, if well tolerated. 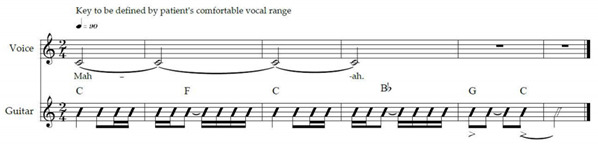 Note: In case of excessive perceived nasality, participant may be advised to “adopt a pleasant facial expression”, slightly lifting the zygomatic muscles (“lifting the cheeks”).
**XII. Laryngeal Elevation through Vocalisation (Gliding Vowels) Exercise**
**Time** (approximately): 5 min**Therapy objective**: To facilitate sustained laryngeal elevation.**Materials and equipment**: Acoustic guitar; metronome (Android smartphone, Metronome Beats application for Android, Kinivo ZX120 Mini Portable Wired Speaker for amplification).**Procedure**: MT models the exercise and sings together with the participant. A comfortable tone from the participant’s midrange is chosen for the exercise. Starting from this tone on “mah” syllable, participant is instructed to slide up a major third interval by beat 4 (the second dotted quarter) of the measure and to sustain this tone until the end of the measure. This singing pattern is repeated for 3 more measures, as the harmony changes. Then, 4 measures on “mee” and then 4 measures on “moo” follow. If well tolerated, the whole exercise may be repeated once and a major third interval may be increased to a perfect fifth for increased laryngeal elevation. Live guitar accompaniment is provided by MT, in 12/8, 130 bpm, one harmony per measure, with accompanying audible metronome click. Repeat 2 times if well tolerated. 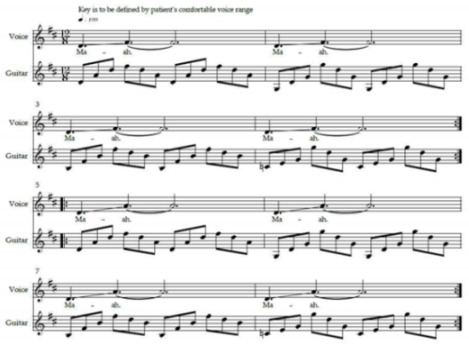
**XIII. Vocal Cord Relaxation Exercises**
**Time** (approximately): 2 min.**Therapy objective**: To relax vocal cords following the singing exercises.**Materials and equipment**: None.**Procedure**: Participant is instructed to use diaphragmatic or mixed types of breathing during these relaxation exercises. XIII.1. Vocal fry exercise Participant is instructed to take a deep breath and to make a vocal fry sound for the duration of the exhale. Can be repeated.XIII.2. Deep breathing (3 repetitions)Participant is instructed to breathe in through the nose, with the mouth slightly open, then breathe out through the mouth, without sound. Air enters and escapes from the lungs with no effort: it is not forced in any way. Participant can choose to use diaphragmatic or mixed (diaphragmatic and chest) types of breathing. XIII.3. Exhale on hard [h] (3 repetitions)Participant is instructed to continue deep breathing and then to exhale on a “lazy”, long, uninterrupted hard [h] sound.
**XIV. Preferred Song Performance**
**Time** (approximately): 5 min.**Therapy objective**: To relax vocal cords following the singing exercises. To reinforce all the voice skills practiced in previous exercises (body alignment and posture, diaphragmatic breathing, proper phonation, soft palate elevation, relaxed consonant articulation, etc.), to provide a motivating reward at the end of the session. **Materials and equipment**: Acoustic guitar. Optional: song lyrics printout.**Procedure**: Participant is invited to sing his/her preferred song in a comfortable range, at a comfortable tempo, with live guitar accompaniment provided by MT. MT may choose to sing together with participant to encourage participation and to model healthy singing technique. Both the participant and the MT provide brief feedback on the performance at the end of the song. Selection of the song for this exercise will occur as follows. At recruitment, each participant will be asked to provide the list of his/her 3 to 5 favourite songs to sing. MT will choose one of these songs to include into the protocol, giving preference to songs with simple melodic and harmonic structure, enough pauses for proper phrasing, moderate tempo, and emotionally neutral or “positive”.
**XV. Session Closure and Assessment**
**Time** (approximately): 5 min.**Therapy objective**: Assessment. **Materials and equipment**: NRS (respiration) and NRS (voice) sheets; NRS data sheet; notebook for note taking.**Procedure**: MT closes the session, acknowledges participant’s effort and reminds of the upcoming session(s) if any are left. The following information is gathered and recorded into the patient’s individual data sheet: (1) VAS for current perceived ease of respiration, and (2) VAS for current perceived ease of voice production. MT instructs (or reminds) the participant to practice recommended daily exercises, if possible, always taking precautions to avoid strain and exhaustion (to “stop if something doesn’t feel right”). MT instructs (or reminds) the participant to follow the voice health guidelines [79].

**Table 2 brainsci-12-00494-t002:** Outcome measures and data collection summary table.

Outcome Measure	Outcome Measure Description	Data Collection Points
**FEASIBILITY OUTCOME MEASURES**		
Recruitment	All newly diagnosed patients at ALS Moscow Centre living within the Moscow city limits and meeting the inclusion and exclusion criteria were invited to participate, until the desired sample size (*n* = 8) was achieved or the cut-off recruitment date. Target recruitment over 80% was considered the marker of a successful feasibility trial.	Prior to the cut-off recruitment date
Retention	The total number of participants recruited was compared with the total number of participants who completed this study. A retention rate over 70% at the end of follow-up period was considered the marker of a successful feasibility trial.	Week 1, week 16
Adherence	The number of music therapy sessions attended by each participant was recorded as an adherence measure for this study. Mean adherence across all participants who completed this study was calculated. Mean adherence for the group calculated at over 75% music therapy sessions delivered was considered the marker of a successful feasibility trial.	Each music therapy session (total of 12), throughout the therapy phase (weeks 6–12)
Tolerance	Short-term tolerability of the music therapy treatment protocol was assessed by measuring change in ratings on the self-reported Ease of Respiration Numeric Rating Scale and change in ratings on the self-reported Ease of Speech Numeric Rating Scale before and after every music therapy session. The outcome could range from “1” (very difficult) to “10” (very easy). Recorded by the music therapist.	Before and after each music therapy session (total of 12), throughout the therapy phase (weeks 6–12)
Self-motivation	Self-reported adherence to a suggested independent exercise routine was recorded to assess levels of self-motivation that participants demonstrated with regard to music therapy treatment. Any attempt to practice was recorded. The number of independent exercises sets performed, in proportion to the number of days when no visit from the music therapist was scheduled, recorded during the sessions from 4 to 12 during the 6 week treatment phase, was calculated, measured in percent.	Each music therapy session starting at session 4 (total of 9), throughout the therapy phase (weeks 7–12)
Treatment experience of participants—persons with ALS	Semi-structured interviews with research participants—persons with ALS—were conducted pre-treatment (week 5) and at the end of the follow-up period (week 16). Participant’s answers to open questions in regard to expectations for and impressions of music therapy treatment were written down during a home visit by a trained research assistant. Interpretative phenomenological analysis was applied to find prominent common themes across the semi-structured interviews.	Week 5, week 16
Caregiver’s experience	Semi-structured interviews with caregivers were conducted prior to treatment (week 5) and at the end of the follow-up period (week 16). Primary caregivers, if identified by participants—persons with ALS—and only with their permission, were approached for the interviews. Caregivers’ answers to open questions in regard to expectations for and impressions of music therapy treatment were written down during a home visit by a trained research assistant. Interpretative phenomenological analysis was applied to find prominent common themes across the semi-structured interviews.	Week 5, week 16
Music therapist’s perspective	Individual treatment notes taken after each session and generalised “field notes” taken twice a week, after completion of each consequent session by all the participants, were submitted by the music therapist. Thematic narrative analysis of the notes was conducted.	Each music therapy session (total of 12), throughout the therapy phase (weeks 7–12)
	**BIOMEDICAL OUTCOME MEASURES**	
**Outcome measures to assess the long-term changes in respiration**		
Change of Forced Vital Capacity (FVC) from baseline at Week 6, week 12, week 16	Forced Vital Capacity (FVC) is a standard spirometry test which measures the volume of air that can forcibly be blown out after full inspiration; measured in %. Measured during a home visit by a nurse.	Week 1, week 6, week 12, week 16
Change of Maximal Inspiratory Pressure (MIP) from baseline at week 6, week 12, week 16	Maximal Inspiratory Pressure (MIP) is the inspiratory pressure generated against a completely occluded airway; used to evaluate inspiratory respiratory muscle strength; measured in cm H2O. Measured during a home visit by a nurse.	Week 1, week 6, week 12, week 16
Change of Maximal Expiratory Pressure (MEP) from baseline at week 6, week 12, week 16	Maximal Expiratory Pressure (MEP) is a measure of the strength of respiratory muscles, obtained by having the patient exhale as strongly as possible against a mouthpiece; measured in cm H2O. Measured during a home visit by a nurse.	Week 1, week 6, week 12, week 16
**Outcome measures to assess the long-term changes in cough**		
Change of Peak Expiratory Flow (PEF) from baseline at week 6, week 12, week 16	Peak Expiratory Flow (PEF) is a measure of cough effectiveness, portable peak flow meter was used; measured in %. Measured during a home visit by a nurse.	Week 1, week 6, week 12, week 16
**Outcome measures to assess the long-term changes in speech**		
The Change of Center for Neurologic Study Bulbar Function Scale (CNS-BFS) Speech subscore from baseline at week 6, week 12, week 16	The Center for Neurologic Study Bulbar Function Scale (CNS-BFS) consists of three domains (swallowing, speech, and salivation), each of which is assessed with a 7-item, self-report questionnaire. Each question is scored from “1” (does not apply) to “5” (applies most of the time). Speech domain subscore can range from “7” (best outcome) to “35” (worst outcome). The result was recorded during a home visit by a trained research assistant.	Week 1, week 6, week 12, week 16
Change in acoustic assessment parameters of recorded voice from baseline at week 6, week 12, week 16	Voice samples were recorded during a home visit by a trained research assistant digitally in .wav format, using a Shure WH20XLR Dynamic Headset Microphone, Alesis iO Dock audio interface, Apple iPad 2 tablet and GarageBand software. Acoustic analysis of the voice samples was conducted using the PRAAT linguistic computer program to calculate the following outcome measures: Maximum Phonation Time (MPT), sound /a/, measured in seconds; Maximum Repetition Rate—Alternating (AMR), /pataka/ sequence, measured in total number of syllables uttered as fast and as clear as possible on one breath; Maximum Repetition Rate—Sequential (SMR), /ba/ syllable, measured in total number of syllables uttered as fast and as clear as possible on one breath; Jitter, local, sound /a/, measured in percent, Shimmer, local, sound /a/, measured in percent; Harmonics-to-Noise Ratio (HNR), measured in Db, sustained /a/; Vowel Space Area (VSA), separate vowels /a, e, i, o, u/, measured in squared Hz; Fundamental frequency (F0), oral reading, measured in Hz; Speaking rate, oral reading, measured in words per minute; Speech–pause ratio, oral reading, measured in seconds per minute; Pause frequency, oral reading, measured in number of pauses per minute; Fundamental frequency (F0), spontaneous speech, measured in Hz; Speaking rate, spontaneous speech, measured in words per minute; Speech–pause ratio, spontaneous speech, measured in seconds per minute; Pause frequency spontaneous speech, measured in number of pauses per minute.	Week 1, week 6, week 12, week 16
Change in perceptual assessment parameters of recorded voice from baseline at week 6, week 12, week 16	Voice samples were recorded during a home visit by a trained research assistant digitally in .wav format, using a Shure WH20XLR Dynamic Headset Microphone, Alesis iO Dock audio interface, Apple iPad 2 tablet and GarageBand software. Perceptual analysis of the voice samples was performed by three qualified speech-language specialists to assess change in hypernasality level of spontaneous speech, measured in points. Inter-rater reliability was calculated for the perceptual analysis results.	Week 1, week 6, week 12, week 16
**Outcome measures to assess the long-term changes in swallowing**		
The Change of Center for Neurologic Study Bulbar Function Scale (CNS-BFS) Swallowing subscore from baseline at week 6, week 12, week 16	The Center for Neurologic Study Bulbar Function Scale (CNS-BFS) consists of three domains (swallowing, speech, and salivation), each of which is assessed with a 7-item, self-report questionnaire. Each question is scored from “1” (does not apply) to “5” (applies most of the time). Swallowing domain subscore can range from “7” (best outcome) to “35” (worst outcome). The result was recorded during a home visit by a trained research assistant.	Week 1, week 6, week 12, week 16
Change in videofluoroscopic swallowing study (VFSS) results from baseline at week 6, week 12	VFSS (videofluoroscopic swallowing study), an X-ray-based method of evaluating a person’s swallowing ability, was performed by a trained specialist during a visit to a laboratory using a BV Pulsera Mobile C-arm fluoroscope, pulsing at 30 pulses per second and recorded on built-in Medical DVD Recorder at 30 frames per second. Each participant swallows 10 mL of nectar and pudding-thick liquid boluses, thickened with a xanthan gum-based thickener (i.e., Nestle Thicken-Up Clear^®^) and mixed to 40% weight-to-volume concentration with BarVIPS powder. VFSS video clips were reviewed and scored by a trained speech-language pathologist, using frame-by-frame analysis following operational definitions outlined by [124]. The following outcome parameters were calculated from VFSS video clips, recorded at three time points, to assess long-term changes in swallowing: Time-to-Laryngeal Vestibule Closure, nectar 10 mL, measured in ms; Time-to-Laryngeal Vestibule Closure, pudding 10 mL, measured in ms; Maximum Pharyngeal Constriction Area, nectar 10 mL, measured in % C2–42; Maximum Pharyngeal Constriction Area, pudding 10 mL, measured in % C2–42; Peak Position of the Hyoid Bone, nectar 10 mL, measured in % C2–4, Peak Position of the Hyoid Bone, pudding 10 mL, measured in % C2–4; Penetration–Aspiration Scale Score (worst), nectar 10 mL, measured in points; Penetration–Aspiration Scale Score (worst), pudding 10 mL, measured in points; Total Pharyngeal Residue C24 area, nectar 10 mL, measured in % C2–4; Total Pharyngeal Residue C24 area, pudding 10 mL, measured in % C2–4; Laryngeal vestibule closure, nectar 10 mL, described as complete, partial, or incomplete; Laryngeal vestibule closure, pudding 10 mL, described as complete, partial, or incomplete.	Week 1, week 6, week 12

**Table 3 brainsci-12-00494-t003:** FVC (%) individual and mean measurements at four assessment time points.

Participant	FVC_t1	FVC_t2	FVC_t3	FVC_t4
1	76	75	89	80
2	69	83	85	79
3	121	115	113	113
4	87	83	88	72
5	63	60	61	63
6	82	73	60	64
7	84	63	62	83
Mean	83.143	78.857	79.714	79.143

**Table 4 brainsci-12-00494-t004:** MIP (cm H_2_O) individual and mean measurements at four assessment time points.

Participant	MIP_t1	MIP_t2	MIP_t3	MIP_t4
1	73	69	91	98
2	44	26	57	81
3	41	38	41	42
4	55	62	49	54
5	32	39	33	42
6	70	56	60	58
7	48	40	57	51
Mean	51.857	47.143	55.429	60.857

**Table 5 brainsci-12-00494-t005:** MEP (cm H_2_O) individual and mean measurements at four assessment time points.

Participant	MEP_tp1	MEP_tp2	MEP_tp3	MEP_tp4
1	68	65	94	79
2	63	76	83	87
3	67	81	80	77
4	91	86	103	91
5	35	47	35	34
6	79	59	57	54
7	55	35	50	51
Mean	65.429	64.143	71.714	67.571

**Table 6 brainsci-12-00494-t006:** PEF (%) individual and mean measurements at four assessment time points.

Participant	PCF_t1	PCF_t2	PCF_t3	PCF_t4
1	30	22	36	82
2	120	103	121	128
3	99	106	107	127
4	109	94	100	93
5	66	50	69	66
6	68	68	75	75
7	64	75	80	80
Mean	79.429	74.000	84.000	93.000

**Table 7 brainsci-12-00494-t007:** CNS-BFS speech subscore (points) individual and mean measurements at four assessment time points.

Participant	CNSBFSsp_t1	CNSBFSsp_t2	CNSBFSsp_t3	CNSBFSsp_t4
1	18	16	14	14
2	8	13	8	10
3	8	10	10	9
4	8	7	10	9
5	24	27	24	29
6	12	13	15	16
7	10	9	9	9
Mean	12.571	13.571	12.857	13.714

**Table 8 brainsci-12-00494-t008:** MPT (seconds) individual and mean measurements at four assessment time points.

Participant	MPT_t1	MPT_t2	MPT_t3	MPT_t4
1	15.96	4.66	12.23	6.95
2	19.66	4.62	13.46	8.14
3	8.52	7.13	6.25	8.13
4	11.84	15.7	14.48	11.51
5	6.68	8.48	9.26	9.48
6	17.1	15.3	22.37	18.04
7	3.61	4.34	7.4	3.71
Mean	11.910	8.604	12.207	9.423

**Table 9 brainsci-12-00494-t009:** Jitter (local, %) individual and mean measurements at four assessment time points.

Participant	Jitter_t1	Jitter_t2	Jitter_t3	Jitter_t4
1	0.431	0.302	0.172	0.376
2	0.318	0.609	0.187	0.519
3	0.371	0.377	0.547	0.378
4	0.46	0.701	0.254	0.356
5	0.308	0.366	0.33	0.314
6	0.422	0.597	0.389	0.497
7	0.537	0.306	0.512	0.436
Mean	0.407	0.465	0.342	0.411

**Table 10 brainsci-12-00494-t010:** Shimmer (local, %) individual and mean measurements at four assessment time points.

Participant	Shimmer_t1	Shimmer_t2	Shimmer_t3	Shimmer_t4
1	2.252	3.479	1.031	2.755
2	5.784	3.994	1.275	5.758
3	2.994	2.68	4.526	2.827
4	3.553	3.262	3.225	6.961
5	1.983	2.505	1.285	2.034
6	3.799	8.194	5.643	8.031
7	4.133	2.326	5.545	2.386
Mean	3.500	3.777	3.219	4.393

**Table 11 brainsci-12-00494-t011:** HNR (Db) individual and mean measurements at four assessment time points.

Participant	NHR_t1	NHR_t2	NHR_t3	NHR_t4
1	18.454	21.141	25.74	21.167
2	21.138	19.715	30.726	18.556
3	20.314	22.39	18.359	19.493
4	20.223	22.081	21.812	17.79
5	23.402	25.091	25.499	23.161
6	20.667	15.408	18.941	15.251
7	18.395	21.538	19.413	20.543
Mean	20.370	21.052	22.927	19.423

**Table 12 brainsci-12-00494-t012:** MRR-A (syllables, total) individual and mean measurements at four assessment time points.

Participant	MRR-A_t1	MRR-A_t2	MRR-A_t3	MRR-A_t4
1	33	33	54	43
2	171	69	108	60
3	108	102	126	93
4	48	87	117	170
5	12	3	21	15
6	66	51	45	45
7	36	30	45	36
Mean	67.714	53.571	73.714	66.000

**Table 13 brainsci-12-00494-t013:** MRR-S (syllables, total) individual and mean measurements at four assessment time points.

Participant	MRR-S_t1	MRR-S_t2	MRR-S_t3	MRR-S_t4
1	32	29	29	30
2	77	27	81	57
3	89	91	105	101
4	44	20	92	86
5	11	13	21	14
6	39	48	43	47
7	13	16	45	27
Mean	43.571	34.857	59.429	51.714

**Table 14 brainsci-12-00494-t014:** VSA (Hertz, squared) individual and mean measurements at four assessment time points.

Participant	VSA_t1	VSA_t2	VSA_t3	VSA_t4
1	335,474	383,408.5	293,932.5	399,866.5
2	194,705	689,702.5	615,575	774,734.5
3	435,659	478,926.5	394,459	422,385
4	422,040.5	469,824.5	479,047.5	431,202
5	287,983.5	116,180	340,084.5	398,212
6	208,821	226,215	190,827	199,621
7	430,167	242,673.5	433,381	309,478
Mean	330,692.9	372,418.6	392,472.4	419,357.0

**Table 15 brainsci-12-00494-t015:** Fundamental frequency (Hz) individual and mean measurements at four assessment time points.

Participant	Freq_t1	Freq_t2	Freq_t3	Freq_t4
1	162.2	161.8	162.6	179.9
2	232.6	243	231.2	234.9
3	161.1	156.2	177	176.5
4	127.2	129.3	145.7	130.6
5	228.7	224.6	212.8	213.3
6	120.3	143.8	139.4	157.4
7	193.7	194.5	196.2	187.5
Mean	175.114	179.029	180.700	182.871

**Table 16 brainsci-12-00494-t016:** Speaking rate (words per minute) individual and mean measurements at four assessment time points.

Participant	SR_t1	SR_t2	SR_t3	SR_t4
1	68	60	72	72
2	130	142	136	136
3	138	110	130	114
4	98	106	110	110
5	72	62	52	54
6	84	74	76	76
7	98	76	88	88
Mean	98.286	90.000	94.857	92.857

**Table 17 brainsci-12-00494-t017:** Speech–pause ratio (seconds per minute) individual and mean measurements at four assessment time points.

Participant	SPAt_t1	SPAt_t2	SPAt_t3	SPAt_t4
1	21	26	17	14
2	9	8	10	10
3	3	8	4	5
4	13	13	11	10
5	15	18	17	15
6	11	11	8	8
7	10	15	12	13
Mean	11.714	14.143	11.286	10.714

**Table 18 brainsci-12-00494-t018:** Hypernasality level (points) individual and mean measurements at four assessment time points.

Participant	Nasality_tp1	Nasality_tp2	Nasality_tp3	Nasality_tp4
1	1.33	1.67	2.33	2.00
2	4.00	4.00	4.00	3.67
3	3.67	3.00	3.33	3.33
4	4.00	3.67	4.00	3.67
5	1.67	1.67	1.00	1.00
6	3.00	2.67	2.00	2.00
7	2.67	2.33	3.00	2.33
Mean	2.905	2.714	2.810	2.571

**Table 19 brainsci-12-00494-t019:** CNS-BFS swallowing subscore (points) individual and mean measurements at four assessment time points.

Participant	CNSBFSsw_t1	CNSBFSsw_t2	CNSBFSsw_t3	CNSBFSsw_t4
1	9	7	8	8
2	11	13	10	10
3	11	14	8	7
4	13	12	10	13
5	16	19	18	19
6	13	16	14	20
7	12	12	12	12
Mean	12.143	13.286	11.429	12.714

**Table 20 brainsci-12-00494-t020:** LVCrt, nectar 10 mL (ms) individual and mean measurements at three assessment time point.

Participant	LVCrtn_t1	LVCrtn_t2	LVCrtn_t3
2	360	400	240
3	240	160	240
4	240	280	280
5	600	440	440
6	440	400	160
Mean	376	336	272

**Table 21 brainsci-12-00494-t021:** LVCrt, pudding 10 mL (ms) individual and mean measurements at three assessment time points.

Participant	LVCrtp_t1	LVCrtp_t2	LVCrtp_t3
2	400	240	240
3	360	360	240
4	200	240	120
5	640	360	400
6	280	440	320
Mean	376	328	264

**Table 22 brainsci-12-00494-t022:** MPCAn, nectar 10 mL (% C2–42) individual and mean measurements at three assessment time points.

Participant	MPCAnn_t1	MPCAnn_t2	MPCAnn_t3
2	3.40941758	10.13944761	2.543253
3	2.9455915	5.710712639	7.754257
4	3.51707662	5.445806512	2.597211
5	9.64075675	6.663143871	3.893648
6	11.639785	14.51433228	15.84521
Mean	6.23052549	8.494688581	6.526715

**Table 23 brainsci-12-00494-t023:** MPCAn, pudding 10 mL (%C2–42) individual and mean measurements at three assessment time points.

Participant	MPCAnp_t1	MPCAnp_t2	MPCAnp_t3
2	3.079057604	4.330900777	4.025907331
3	4.67484866	7.200453733	6.510919871
4	5.433459744	8.731734229	3.957121688
5	7.909192227	10.0939909	6.568875036
6	16.31579166	40.65572656	33.12445659
Mean	7.482469979	14.20256124	10.8374561

**Table 24 brainsci-12-00494-t024:** PeakXY, nectar 10 mL (%C2–4) individual and mean measurements at three assessment time points.

Participant	PeakXYn_t1	PeakXYn_t2	PeakXYn_t3
2	149.1	152.3	156.9
3	150.9	141.2	145.5
4	180.6	184.3	183.9
5	140.9	148	146.2
6	163.9	160.6	160.5
Mean	157.08	157.28	158.6

**Table 25 brainsci-12-00494-t025:** PeakXY, pudding 10 mL (%C2–4) individual and mean measurements at three assessment time points.

Participant	PeakXYp_t1	PeakXYp_t2	PeakXYp_t3
2	151.5	145.9	149.7
3	148.9	151.1	148.7
4	196.2	181.8	199.9
5	145.3	152.1	148.2
6	172	148.7	165.2
Mean	162.78	155.92	162.34

**Table 26 brainsci-12-00494-t026:** PAS (worst), nectar 10 mL (points) individual and mean measurements at three assessment time points.

Participant	PASn_t1	PASn_t2	PASn_t3
2	1	1	5
3	1	1	2
4	1	1	2
5	2	6	5
6	1	1	1
Mean	1.2	2	3

**Table 27 brainsci-12-00494-t027:** PAS (worst), pudding 10 mL (points) individual and mean measurements at three assessment time points.

Participant	PASp_t1	PASp_t2	PASp_t3
2	5	1	1
3	1	2	2
4	1	1	1
5	1	5	3
6	1	1	3
Mean	1.8	2	2

**Table 28 brainsci-12-00494-t028:** TotC24, nectar 10 mL (%C2–4) individual and mean measurements at three assessment time points.

Participant	totC24n_t1	totC24n_t2	totC24n_t3
2	1.5114466	6.719501338	1.381531396
3	1.4108957	3.142448902	4.130194898
4	4.8939937	2.953970978	2.907582467
5	8.1621392	4.232066181	1.982200267
6	13.582741	28.79061578	18.01529424
Mean	5.9122432	9.167720635	5.683360654

**Table 29 brainsci-12-00494-t029:** TotC24, pudding 10 mL (%C2–4) individual and mean measurements at three assessment time points.

Participant	totC24p_t1	totC24p_t2	totC24p_t3
2	1.67565221	4.61141829	3.509672232
3	6.00775756	8.55609461	5.723845175
4	2.91294337	7.47843618	3.484580659
5	3.55738697	6.66928515	4.553890179
6	17.9090241	45.1455462	21.13985385
Mean	6.41255284	14.4921561	7.682368419

**Table 30 brainsci-12-00494-t030:** LVC, nectar 10 mL, individual descriptions at three assessment time points.

Participant	LVCn_t1	LVCn_t2	LVCn_t3
2	Partial	complete	partial
3	Complete	complete	complete
4	Compete	complete	complete
5	Partial	complete	partial
6	Complete	complete	complete

**Table 31 brainsci-12-00494-t031:** LVC, pudding 10 mL, individual descriptions at three assessment time points.

Participant	LVCp_t1	LVCp_t2	LVCp_t3
2	Partial	complete	partial
3	Complete	complete	complete
4	Complete	complete	complete
5	Complete	partial	partial
6	Complete	complete	complete

**Table 32 brainsci-12-00494-t032:** Suggested biomedical outcome measures and parameters for the efficacy study.

**Outcome Measures to Assess the Long-Term Changes in Respiration**		
Change of Forced Vital Capacity (FVC) Change of Maximal Inspiratory Pressure (MIP) Change of Maximal Expiratory Pressure (MEP)		
**Outcome measures to assess the long-term changes in cough**		
	Peak Expiratory Flow (PEF)	
**Outcome measures to assess the long-term changes in speech**		
Change of the Center for Neurologic Study Bulbar Function Scale (CNS-BFS) Speech subscore		
Change in acoustic assessment parameters of recorded voice samples: Maximum Phonation Time (MPT) Maximum Repetition Rate—Alternating (AMR) Maximum Repetition Rate—Sequential (SMR) Jitter, local Shimmer, local Harmonics-toNoise Ratio (HNR) Vowel Space Area (VSA) Fundamental frequency (F0) Speaking rate, oral reading Speech–pause ratio, oral reading Pause frequency, oral reading		
Change of aerodynamic measure of nasal flow		
**Outcome measures to assess the long-term changes in swallowing**		
Change of the Center for Neurologic Study Bulbar Function Scale (CNS-BFS) Swallowing subscore		
Change in videofluoroscopic swallowing study (VFSS): Time-to-Laryngeal Vestibule Closure, liquid 10 mL, nectar 10 mL, pudding 10 mL Maximum Pharyngeal Constriction Area, liquid 10 mL, nectar 10 mL, pudding 10 mL Maximum Pharyngeal Constriction Area, liquid 10 mL, nectar 10 mL, pudding 10 mL Peak Position of the Hyoid Bone, liquid 10 mL, nectar 10 mL, pudding 10 mL Penetration–Aspiration Scale Score (worst), liquid 10 mL, nectar 10 mL, pudding 10 mL Total Pharyngeal Residue C24area, liquid 10 mL, nectar 10 mL, pudding 10 mL Laryngeal vestibule closure, liquid 10 mL, nectar 10 mL, pudding 10 mL		

## Data Availability

Not applicable.

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
