# Peer review of "Home-Based Music Therapy to Support Bulbar and Respiratory Functions of Persons with Early and Mid-Stage Amyotrophic Lateral Sclerosis—Protocol and Results from a Feasibility Study"

_brainsci, 2022, doi:10.3390/brainsci12040494_

Round 1
Reviewer 1 Report
Amyotrophic lateral sclerosis (ALS) is a neurodegenerative disease, and there is no cure. Given that the current medications, riluzole and edaravone, only provide minor advantages for ALS patients, supporting rehabilitations such as music therapy are critical to improve patients' quality of life. This work by Alisa et.al reported a detailed study protocol of music therapy to support bulbar and respiratory functions of persons with early and mid-stage ALS, which has great benefits for ALS patients. Overall, I believe this is a well-written manuscript that is publishable. The only question I have is whether this registered study has been completed. If it's done, the authors may want to provide the results in this publication to help interpret the protocol better.
Author Response
Dear reviewer (1), Thank you for your time and thought. We have responded to your comments and done as you suggested, including study findings, discussion and conclusion. Please see attached revised version, with track changes as per journal guidelines.
Many thanks.

Reviewer 2 Report
This is a very interesting and novel study to examine music therapy and mild exercise on outcomes in ALS, particularly for breathing, speech, and swallowing. The protocols developed are clear, specific, and repeatable. The study is nice designed and all supporting documentation was clearly presented in the Appendix. The Introduction and Methods were great.
The key issue is that there is not Results section. I had to scroll through multiple times to make sure I hadn't missed something. It goes from Methods to a short Conclusion paragraph. No quantitative results are reported. The authors state that are not using statistical significance but rather trends. Yet, no results figures or trend data is shown. Even if there are no meaningful results yet to report (due to number of enrolled patients, stage of the clinical trial, etc.), the authors still need to write an intended analysis plan.
If the point of this paper is to only provide a possible protocol to be examined later, this needs to be better clarified in the abstract and Introduction. If the point of the paper is more to illustrate preliminary results, even if the sample size of patients is still too small for significance, this should also be clarified.
Even if the key point is to only provide a protocol that is to be tested, the paper still needs a clear analytical plan to insure repeatability. A power analysis, an expected standard deviation, a way to measure a trend...something. There are many ways this could be done, but not a single one was presented in full. If this method and corresponding analysis plan is to be adopted by others, a cohesive analysis plan is expected. Ideas could include an overall analysis combining all metrics using unsupervised clustering, random forest feature weights, Kendall tau ranking, etc. If it's only to look at specific quantitative measures the author's listed, than specific statistical tests could be suggested and a power analysis performed based on existing literature data for expected standard deviation of the metrics given the study's stated inclusion criteria. If this is just about "trends", then trend needs to be further defined and quantified.
Overall, this is a wonderful idea and well thought out method. It also seeks to address a key question in ALS management. However, the authors need to write a coherent analysis plan.
Author Response
Dear reviewer (2), Thank you for your time and thought. We have inserted the detail you have requested- analysis methods (p. 27-28), study findings following this and then discussion and conclusion. Please see attached article with track changes-as per the journal submission guidelines

Round 2
Reviewer 2 Report
The authors have added the requested analysis, which greatly increases the technical soundness of the work and its interest to the larger ALS community. Recommend only minor spell check and proofreading.
P.S. There are many tables in here, which is mostly unavoidable. A minor suggestion would be to panel tables into groups or related tables 2A, 2B, etc. for reader accessibility in finding tables with similar categorical metrics. However, this is not required and would be completely at the discretion of the journal formatting team.